# Activated-memory T cells influence naïve T cell fate: a noncytotoxic function of human CD8 T cells

Kazuki Sasaki[1,2,3,8], Mouhamad Al Moussawy[1,2,8], Khodor I. Abou-Daya [1,2], Camila Macedo[1,2], Amira Hosni-Ahmed[4], Silvia Liu[5,6], Mariam Juya[1,2], Alan F. Zahorchak[1,2], Diana M. Metes[1,2,7], Angus W. Thomson[1,2,5,7], Fadi G. Lakkis[1,2,7] & Hossam A. Abdelsamed [1,2,5✉]

T cells are endowed with the capacity to sense their environment including other T cells around them. They do so to set their numbers and activation thresholds. This form of regulation has been well-studied within a given T cell population – i.e., within the naïve or memory pool; however, less is known about the cross-talk between T cell subsets. Here, we tested whether memory T cells interact with and influence surrounding naïve T cells. We report that human naïve CD8 T cells ($T_N$) undergo phenotypic and transcriptional changes in the presence of autologous activated-memory CD8 T cells ($T_{Mem}$). Following in vitro co-culture with activated central memory cells ($T_{CM}$), ~3% of the $T_N$ acquired activation/ memory canonical markers (CD45RO and CD95) in an MHC-I dependent-fashion. Using scRNA-seq, we also observed that ~3% of the $T_N$ acquired an activated/memory signature, while ~84% developed a unique activated transcriptional profile hybrid between naïve and activated memory. Pseudotime trajectory analysis provided further evidence that $T_N$ with an activated/memory or hybrid phenotype were derived from $T_N$. Our data reveal a non-cytotoxic function of $T_{Mem}$ with potential to activate autologous $T_N$ into the activated/ memory pool. These findings may have implications for host-protection and autoimmunity that arises after vaccination, infection or transplantation.

[1] Department of Surgery, University of Pittsburgh, School of Medicine, Pittsburgh, PA 15213, USA. [2] Starzl Transplantation Institute, University of Pittsburgh, School of Medicine, Pittsburgh, PA 15213, USA. [3] Department of Gastroenterological Surgery, Graduate School of Medicine, Osaka University, Osaka 565-0871, Japan. [4] Department of Chemistry, Division of Biochemistry, Faculty of Science, Fayoum University, Fayoum 63514, Egypt. [5] Pittsburgh Liver Research Center, School of Medicine, Pittsburgh, PA 15213, USA. [6] Department of Pathology, School of Medicine, Pittsburgh, PA 15213, USA. [7] Department of Immunology, University of Pittsburgh, School of Medicine, Pittsburgh, PA 15213, USA. [8]These authors contributed equally: Kazuki Sasaki, Mouhamad Al Moussawy. ✉email: abdelsamedha@upmc.edu

Throughout the lifetime of an organism, biological systems are constantly challenged to regulate their cell numbers, function, and specificity. Along the same lines, the immune system utilizes several mechanisms to continuously sense its environment and respond appropriately. For instance, the pool of mature naïve T cells maintains its survival and function through a low degree of self-reactivity and contact with the common gamma chain cytokine IL-7, while memory T cells slowly proliferate in response to homeostatic cytokines over a long period of time[1–3]. Whether these populations interact and regulate each other is not well understood.

Naïve T cells ($T_N$) continually recirculate between secondary lymphoid tissues and the blood. They dwell in the lymph nodes for 12–24 h before returning to the blood stream via efferent lymphatics[4–6]. Memory T cells patrol nonlymphoid tissues and the spleen but can also circulate and reside in lymph nodes, which is particularly true for the central memory T cell ($T_{CM}$) subset. Upon antigen re-exposure, $T_{CM}$ undergo a rapid transition from a quiescent to a highly activated proliferative/effector state in lymph nodes before migrating to the site of infection to eliminate the pathogen[7,8]. Moreover, a proportion of the activated memory T cells re-enter lymph nodes including those that are antigen-free[9,10]. Despite reports of differential localization of naïve and $T_{CM}$ in the lymph nodes upon infection, substantial overlap between their anatomical locales still exists under steady-state conditions and following activation[11,12]. Hence, it is possible that memory and naïve T cells encounter each other in secondary lymphoid tissues.

Here we hypothesized that upon activation, memory CD8 T cells acquire a noncytotoxic function whereby they interact with naïve CD8 T cells and alter their phenotype and activation state. To test this hypothesis, we designed an in vitro system in which human polyclonal naïve CD8 T cells were co-cultured with autologous, activated memory CD8 T cell subsets. The phenotype and transcriptional profile of naïve cells was analyzed at the end of the co-culture period. We report that in the presence of activated $T_{CM}$ or stem cell memory ($T_{SCM}$) cells, a small proportion (~3%) of autologous naïve CD8 T cells acquired phenotypic and transcriptional features of activated/memory CD8 T cells in an MHC class I-dependent manner, whereas a much larger proportion (~84%) transitioned to a unique activated state, hybrid between naïve and activated memory. These findings imply that activated memory T cells can set the activation threshold of neighboring naïve T cells.

## Results

### Naïve CD8 T cells acquire an activated/memory phenotype in the presence of activated memory CD8 T cells.

To examine the influence of activated memory T cells on neighboring naïve T cells ($T_N$), we sorted human memory CD8 T cells subsets ($T_{EM}$, $T_{CM}$, and $T_{SCM}$) and labeled them with CFSE (Fig. 1a and Suppl. Fig. 1a). We then stimulated each memory subset separately overnight using magnetic beads coated with anti-CD3 and anti-CD28 antibodies. After beads removal, we co-cultured the activated memory CD8 T cell subsets separately with sorted autologous naïve CD8 T cells labeled with cell trace violet (CTV) at a 1:1 ratio without any further exogenous stimulation (Fig. 1a and Suppl. Fig. 1a). As controls, naïve CD8 T cells were cultured alone or in the presence of unstimulated memory CD8 T cells. Seven days later, the phenotype of the naïve CD8 T cells ($CTV^+CFSE^-$) was examined by flow cytometry based on the expression of the lymphoid-tissue homing chemokine receptor CCR7 and the spliced form of tyrosine phosphatase CD45RO (Fig. 1b). In the presence of activated $T_{CM}$ CD8 T cells, we observed a significant (~10-fold) increase in the frequency of naïve CD8 T cells that

acquired an activated/memory phenotype ($CD45RO^+$, ~1.5%) compared to controls (~0.15%) (Fig. 1b, c). Of note, $CD45RO^+$ cells were present in both the $CTV^{hi}$ and the $CTV^{int}$ naïve CD8 T cell population (Suppl. Fig. 1b). Further, we observed a similar change in naïve CD8 T cell phenotype in the presence of activated $T_{SCM}$ CD8 T cells (~1.2%) but to a lesser extent in the presence of activated $T_{EM}$ (~0.5%) (Fig. S1c, d). Since it is expected that $T_{CM}$ CD4 T cells can co-exist with naïve and $T_{CM}$ CD8 T cells in the lymphoid compartment[13], we thought to examine the effect of activated $T_{CM}$ CD4 T cells on naïve CD8 T cells as well. Indeed, we observed a similar effect albeit to a higher extent in the presence of activated $T_{CM}$ CD4 T cells (~8.5%) (Fig. S1e). Ongoing studies are implemented to further understand this phenotype.

We next used advanced imaging flow cytometry to confirm that naïve CD8 T cells acquired an activated/memory phenotype. As shown in Fig. 1d, $CTV^+CFSE^-$ naïve CD8 T cells co-expressed CCR7 and CD45RO after co-culture with activated $T_{CM}$ but not when cultured alone. The low incidence of naïve CD8 T cells that acquired an activated/memory phenotype prompted us to ask whether this phenotype is dependent on the memory:naïve CD8 T cell ratio. We, therefore, repeated the co-culture using a higher ratio of memory to naïve CD8 T cells and found that a 3:1 ratio ($T_{CM}:T_N$) significantly increased the frequency of CD8 T cells that acquired an activated/memory phenotype compared to the 1:1 ratio (~3% vs 1.5%, respectively) (Fig. 1e, f). Additionally, we examined the expression of the Fas receptor (FasR, CD95), which is another activation/memory marker, on the $CTV^+CFSE^-$ cells at the end of the co-culture. We found that the naïve CD8 T cells that acquired an activated/memory phenotype ($CD45RO^+$) upregulated CD95 compared to naïve CD8 T cells that remained $CD45RO^-$ (Fig. 1g, h). Taken together, these results indicate that, in addition to their known cytotoxic functions, activated long-lived memory CD8 T cells ($T_{CM}$ and $T_{SCM}$ subsets) acquire a noncytotoxic function whereby they shift neighboring naïve CD8 T cells towards an activated/memory phenotype.

### Phenotypic and functional analyses of activated memory CD8 T cell subsets.

Memory T cell subsets are known to acquire different proliferation capacities in response to cytokine and/or antigen stimulation[14–19]. Hence, we thought to compare the proliferation of sorted $T_{EM}$, $T_{CM}$, and $T_{SCM}$ CD8 T cells in our in vitro system following overnight stimulation with anti-CD3/CD28 beads (Fig. 2a). As expected, a minority of $T_{EM}$ had divided 7 days post-bead removal (~13% $CFSE^{lo}$) while 95% of $T_{CM}$ and $T_{SCM}$ had proliferated (Fig. 2b). Despite we observed almost 95% of $T_{CM}$ CD8 cell subset proliferated at Day 6 (following removal of beads), the cells underwent gradual proliferation at earlier time point i.e., Day 3 (Fig. S2a). Additionally, all memory subsets downregulated CCR7 following activation (Fig. 2c). These data suggested that once long-lived memory CD8 T cells ($T_{CM}$ and $T_{SCM}$ CD8 T cells) undergo a brief antigen stimulation (18 h), they become committed to proliferation and differentiation programs even in the absence of the original stimulus. Previous studies showed similar phenomenon in mouse naïve and antigen-specific T cells[20–24].

Since we observed that the activated $T_{CM}$ and $T_{SCM}$ subsets had the highest proliferative capacity and the most pronounced effect in our co-culture experiments (Figs. 1c and 2b), we asked whether the acquisition of an activated/memory phenotype by naïve T cells correlates with memory T cell proliferation. To test this hypothesis, we performed simple linear regression test using Pearson correlation coefficient analyses to draw a relationship between two variables: (1) percentage of $CD45RO^+$ $CTV^+$ CD8 T cells and (2) percent dividing memory T cells within the total memory population following co-culture with naïve CD8 T cells (Fig. 2d).

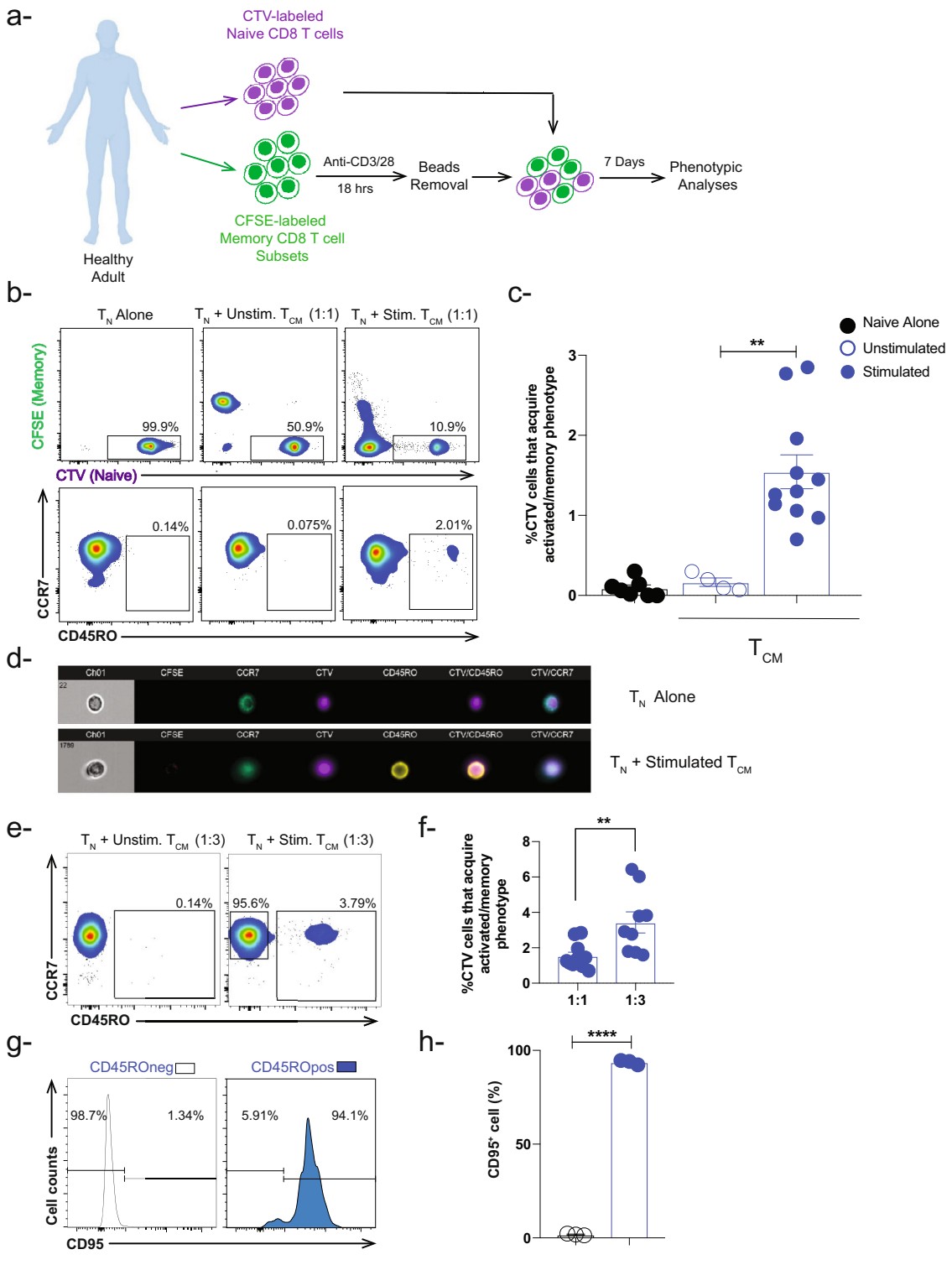

In our analyses, we found a direct correlation between both variables ($R^2 = 0.36$, $p = 0.001$) (Fig. 2e), in which an increase in the proliferation of memory CD8 T cells specially T_CM is associated with increase in the frequency of naïve CD8 T cells with acquired activated/memory phenotype. These data suggest that the differential proliferation capacity of memory T cell subsets (Fig. 2b) could account for their discrepant effect on naïve T cell fate.

**Acquisition of activated/memory phenotype by naïve CD8 T cells is dependent on memory CD8 T cell proliferation**. The direct correlation between the proliferation capacity of memory CD8 T cell subsets and the naïve CD8 T cell phenotype prompted us to ask what will happen if we block the proliferation of activated memory CD8 T cells, could this impair their capacity in influencing the fate of naïve CD8 T cells. To answer this question,

**Fig. 1 Human naïve CD8 T cells undergo phenotypic changes in the presence of autologous activated memory CD8 T cells. a** In vitro co-culture schema showing stimulation of sorted CFSE-labeled human memory CD8 T cell subsets with anti-CD3/CD28 magnetic beads (1:1 ratio). After overnight stimulation, the beads were removed using a plate magnet followed by incubation with CTV-labeled naïve CD8 T cells for 7 days (memory:naive ratio [1:1 ratio and 3:1 ratio]). CTV-labeled cells were examined for activated/memory phenotype. **b** Representative flow cytometry plots depicting gating strategy for live CTV-naïve CD8 T cells expressing CCR7$^{+/-}$ CD45RO$^+$ under three conditions: (T$_N$) alone, naïve co-cultured with unstimulated T$_{CM}$, or with stimulated T$_{CM}$. The gating strategy includes CTV$^{+/int}$ subpopulations and excludes CTV$^{neg}$ cells. This gating strategy has been used in the analyses to exclude CFSE$^{neg}$ activated memory CD8 T cells which might overlap with the CTV$^{neg}$ population. **c** Bar-graph showing percent of CTV-labeled naïve CD8 T cells that acquire activated/memory phenotype based on CCR7 and CD45RO cell surface expression in the presence of stimulated T$_{CM}$ (blue box) at 1:1 ratio (T$_{CM}$:T$_N$). Naïve CD8 T cells were cultured alone or in the presence of unstimulated T$_{CM}$ and were used as controls. Blue closed circles depict stimulation conditions, while open circles depict unstimulated conditions ($n = 4$–11 healthy donors). **d** Representative image ($n = 1$) showing cell surface markers expressed by CTV-labeled naïve CD8 T cells alone and in the presence of stimulated T$_{CM}$ CD8 T cells using advanced imaging flow cytometry. **e** Representative flow cytometry plots and **f** Bar-graph showing the percentage of CTV-labeled naïve CD8 T cells that acquire an activated/memory phenotype in the presence of stimulated T$_{CM}$ at 3:1 ($n = 9$) and 1:1 ($n = 11$) ratio (T$_{CM}$:T$_N$). **g** Histograms showing expression of CD95 (activation/memory marker) in both subpopulations CTV$^+$ CD45RO$^{neg}$ and CD45RO$^+$ in the presence of CFSE-labeled activated T$_{CM}$ CD8 T cells. **h** Bar-graph showing percent of cells expressing CD95 ($n = 3$) **$P < 0.01$ ****$P < 0.0001$. Unpaired nonparametric Mann-Whitney test was used. Data are presented as means ± SEM.

we irradiated activated CFSE-labeled T$_{CM}$ CD8 T cells and co-culture them with CTV-labeled naïve CD8 T cells. As controls, we used activated non-irradiated T$_{CM}$ CD8 T cells. Indeed, upon irradiation, we observed a dramatic reduction in the frequency of naïve with activated/memory phenotype compared to non-irradiated controls (Fig. 3a, b).

To further validate our findings, we used the immunosuppressive drug Cyclosporin A (CsA), a calcineurin inhibitor that suppresses T cell proliferation[25,26]. In this experimental setup, we co-cultured naïve CD8 T cells with activated CFSE-labeled T$_{CM}$ CD8 T cells in the absence or presence of different concentrations of CsA (10 ng/ml and 100 ng/ml). In the presence of CsA, we observed a significant decrease in the proliferation of activated memory T$_{CM}$ CD8 T cells as well as naïve CD8 T cells with acquired activated/memory phenotype compared to co-culture conditions without the drug (Fig. S3a, b). These results demonstrated that the proliferation capacity of activated memory T$_{CM}$ CD8 T cells plays an important role in acquisition of activated/memory phenotype by naïve CD8 T cells.

Since gamma chain cytokines such as IL-2 are crucial in the survival and proliferation of T cells[27,28], while CsA abrogates T cell proliferation through suppression of IL-2 production and other cytokines[29–32], we thought to examine the effect of IL-2 blockade on our phenotype during the co-culture. To achieve such aim, we first determined the levels of IL-2 in the supernatant of activated T$_{CM}$ CD8 T cells following beads stimulation as a guidance for how much anti-IL-2 we should add during the co-culture conditions. As expected, we observed a significant increase in the levels of IL-2 in the supernatant of stimulated T$_{CM}$ CD8 T cells with an average of 1300 pg/ml compared to unstimulated controls (Fig. S3c).

We next activated CFSE-labeled T$_{CM}$ CD8 T cells then co-culture them with CTV-labeled naïve CD8 T cells in the presence of anti-human IL-2 antibody. As controls, we used isotype antibody at the same concentration. Following 6 days of the co-culture, we examined the proliferation capacity of CFSE-labeled T$_{CM}$ CD8 T cells as well as the phenotype of CTV-labeled naïve CD8 T cells. We observed a significant decrease in the proliferation capacity of memory T cells and concurrently the frequency of naïve CD8 T cells with an activated/memory phenotype was much less compared to isotype controls. These data suggest that IL-2 plays an indirect role in the acquisition of activated/memory phenotype by naïve CD8 T cells through induction of memory T cell proliferation (Fig. 3c, d).

However, our approach still did not address whether blocking IL-2 will have a direct effect on naïve CD8 T cells to acquire activated/memory phenotype independent of the proliferation of memory CD8 T cells. Hence, we added rhIL-2 cytokine (1 ng/ml

and 10 ng/ml) to the co-culture conditions of naïve in the presence of unstimulated T$_{CM}$ CD8 T cells for 6 days. In this experimental setup, we did not observe an increase in the frequency of naïve CD8 T cells with acquired activated/memory phenotype as well as proliferation of T$_{CM}$ CD8 T at both concentrations compared to regular co-culture conditions (Fig. 3e, f). These results suggest that MHC-TCR axis (signal 1) and probably other soluble factor(s) could be the early events required to initiate the proliferation of T$_{CM}$ CD8 T cells and hence acquisition of activated/memory phenotype by naïve CD8 T cells. Thus far, our data reveal a cause-and-effect relationship between acquisition of activated/memory phenotype by naïve CD8 T cells and the proliferation capacity of activated memory T$_{CM}$ CD8 T cells.

**Activated memory CD8 T cells alter naïve CD8 T cell phenotype via a contact, MHC class I dependent mechanism.** To further test the above-mentioned hypothesis, we investigated the effect of cell-free supernatant from activated T$_{CM}$ cells on naïve T cells (Fig. 4a). Following 7 days of culture, only ~0.4% of the naïve T cells acquired a CD45RO$^+$ phenotype in the presence of cell-free supernatant compared to ~1.6% of those co-cultured with activated T$_{CM}$ at a 1:1 ratio (Fig. 4c, d). To validate our results, we also used a transwell system in which activated T$_{CM}$ was separated from the naïve cells by a transmembrane during the co-culture period (Fig. 4b). As shown in Fig. 4e, f, CD45RO$^+$ naïve T cells with acquired activated/memory CD8 T cell were scarcely detected compared to contemporaneous control co-culture where the cells were not separated (mean ~0.1% vs 2.2%). These results suggest that contact between naïve and activated memory T cells is necessary for the acquisition of the activated/memory phenotype by naïve cells.

Since the nature of the cell surface molecule that could mediate this phenotype is not known yet, we postulated that contact might be mediated by MHC-I molecules for two reasons: (1) MHC-I molecules are ubiquitously expressed by immune and non-immune cells[33] and (2) MHC-I-self-peptide interaction with TCR plays an important role in lymphopenia induced proliferation (LIP) and the subsequent generation of memory cells from naïve CD8 T cells in the absence of cognate antigen[34,35]. Consequently, we thought first to examine the expression of MHC-I molecule on unstimulated and stimulated T$_{CM}$ cells. Indeed, we observed a significant increase in the mean fluorescence intensity (MFI) of MHC-I following overnight anti-CD3/28 beads stimulation compared to unstimulated controls (Fig. S3d). We next blocked MHC-I molecules by adding a pan anti-MHC-I antibody and examined the phenotype of the naïve CD8 T cells 7 days later (Fig. 5a). We observed a significant decrease in the frequency of

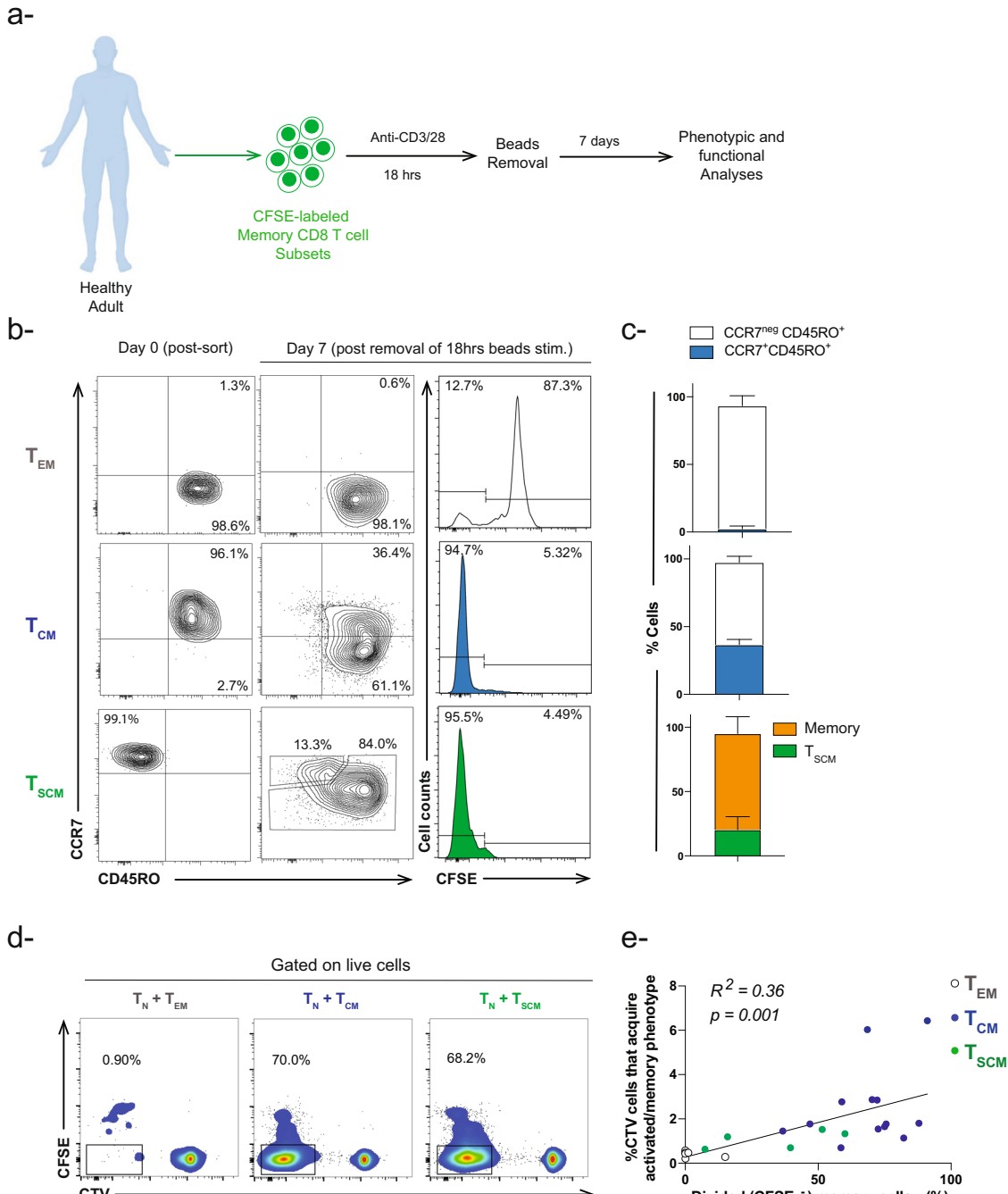

**Fig. 2 Phenotype and proliferation capacity of activated memory CD8 T cell subsets. a** Experimental setup diagram for activation of memory CD8 T cell subsets. **b** Representative flow cytometry contour plots showing the phenotype of memory CD8 T subsets at Day 0 and Day 7 after removal of anti-CD3/CD28 stimulation. Histograms showing proliferation of CFSE-labeled memory CD8 T cell subsets. **c** Bar-graph showing the percentage of memory CD8 T cells that maintain or change their phenotype following removal of TCR stimulation ($n = 3–6$). **d** Representative flow cytometry plots showing proliferation of CFSE-labeled activated memory CD8 T cell subsets in the presence of CTV-labeled naïve CD8 T cells. The percent in the graph represents the proportion of dividing memory CD8 T cells (CFSE^neg). **e** Pearson coefficient correlation analyses between percent of dividing memory CD8 T cell subsets in the presence of naive CD8 T cells versus percent of naïve CD8 T cells that acquired an activated/memory phenotype. Simple linear regression test was applied between the two variables ($n = 24$, $R^2 = 0.36$ $p = 0.001$).

naïve CD8 T cells that acquired an activated/memory phenotype (CD45RO$^+$) upon blocking MHC-I at different concentrations compared to isotype controls (Figs. 5b, c, S3e). It is noteworthy to mention that we observe a change in the frequency of CTV-naïve and CFSE-memory CD8 T cells upon blocking MHC-I compared to controls. To further validate our results, we repeated the co-culture conditions 3:1 (memory:naïve) ratio in the absence or

presence of anti-MHC-I antibody (5 µg/ml). Under these conditions, we observed similar results where upon blocking MHC-I molecules, there was a significant decrease in the frequency of naïve CD8 T cells that acquire activated/memory phenotype (mean ~0.9% vs 3%, respectively) (Fig. 5d, e). In summary, these data indicate that naïve CD8 T cells contact activated memory T cells and acquire an activated/memory

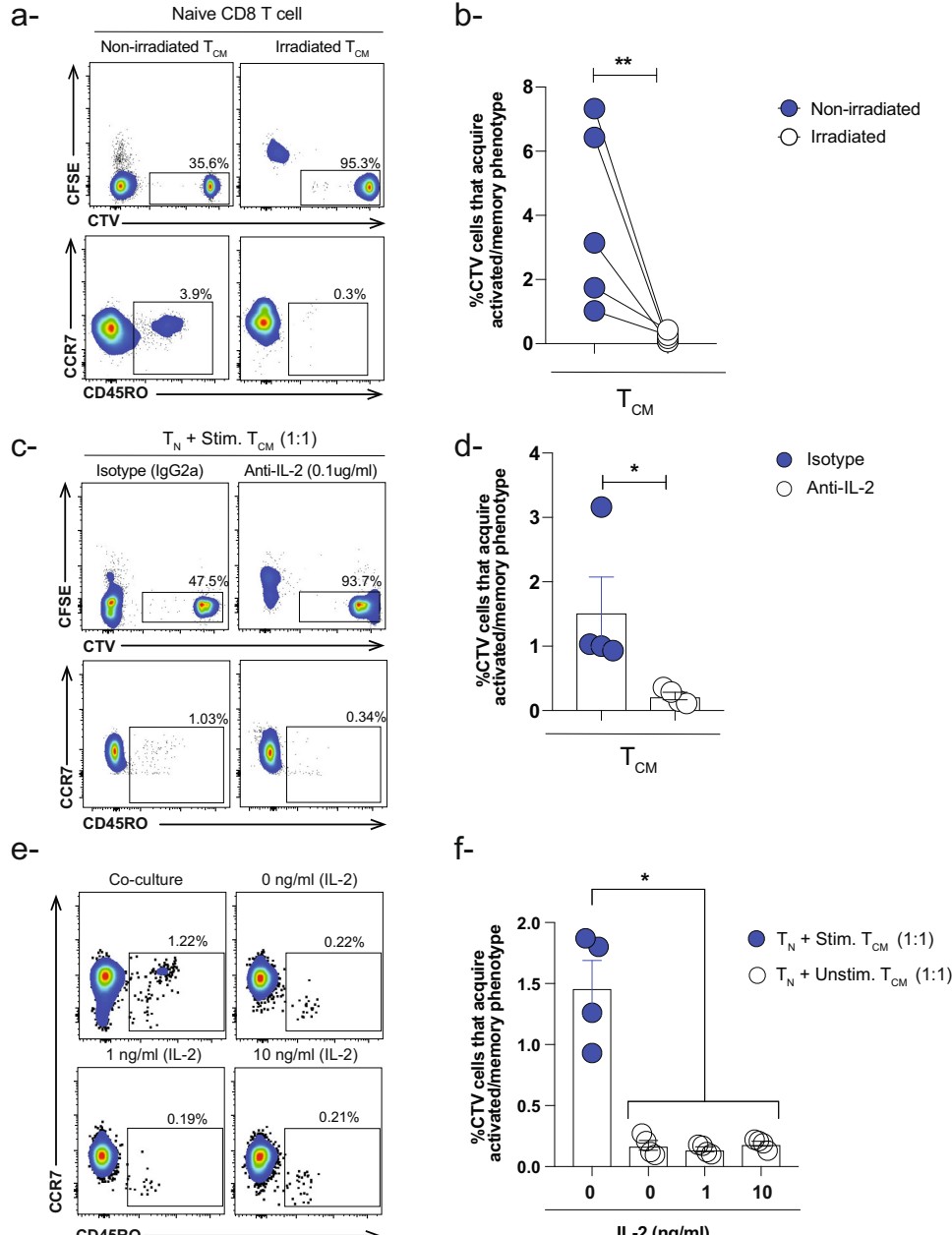

**Fig. 3 Proliferation of activated memory CD8 T cell subsets influence the phenotype of naïve CD8 T cells. a** Representative flow cytometry plots and **b** paired-analysis showing the frequency of naïve CD8 T cells that acquired activated/memory phenotype in the presence of activated non-irradiated and irradiated T$_{CM}$ CD8 T cells (memory:naïve-3:1 ratio). Wilcoxon test (paired nonparametric t-test) was used **$P < 0.01$. Data are presented as means ± SEM ($n = 5$ healthy donors) **c** Representative flow cytometry plots and **d** Bar-graph showing the effect of anti-IL-2 human antibody (0.1 μg/ml) on the frequency of naïve CD8 T cells that acquired activated/memory phenotype in the presence of activated T$_{CM}$ CD8 T cells (1:1) ratio. Isotype IgG2a was used at the same concentration as a control ($n = 4$). Unpaired nonparametric Mann-Whitney test was used. Data are presented as means ± SEM *$P < 0.05$. **e** Representative flow plots and **f** Bar-graph showing the frequency of CTV-labeled naïve CD8 T cells that acquired activated/memory phenotype in the presence of unstimulated T$_{CM}$ CD8 T cells (1:1 ratio) and different concentrations of exogenous rhIL-2 compared to regular co-culture conditions (T$_N$ + stimulated T$_{CM}$ 1:1). Ordinary one-way ANOVA Bonferroni multiple comparison test was used *$P < 0.05$. Data are presented as means ± SEM ($n = 4$).

phenotype via an MHC-I-dependent mechanism albeit gamma chain cytokines could also contribute minimally since we did not observe a complete abrogation of the phenotype.

**Transcriptional changes in naïve CD8 T cells co-cultured with activated memory CD8 T cells**. The identity of an individual T cell cannot be defined solely by its phenotypic features[36]. Hence,

we investigated the nature of these cells based on their transcriptional profile. To identify the transcriptional changes occurring in naïve CD8 T cells co-cultured with activated T$_{CM}$ cells, we sorted the naïve cells at the end of the 7 days co-culture and performed single-cell RNA sequencing (scRNA-seq). As controls, naïve and activated T$_{CM}$ cells were cultured separately for 7 days then single-cell sequenced (Fig. 6a). Uniform Manifold Approximation and Projection (UMAP) dimensionality

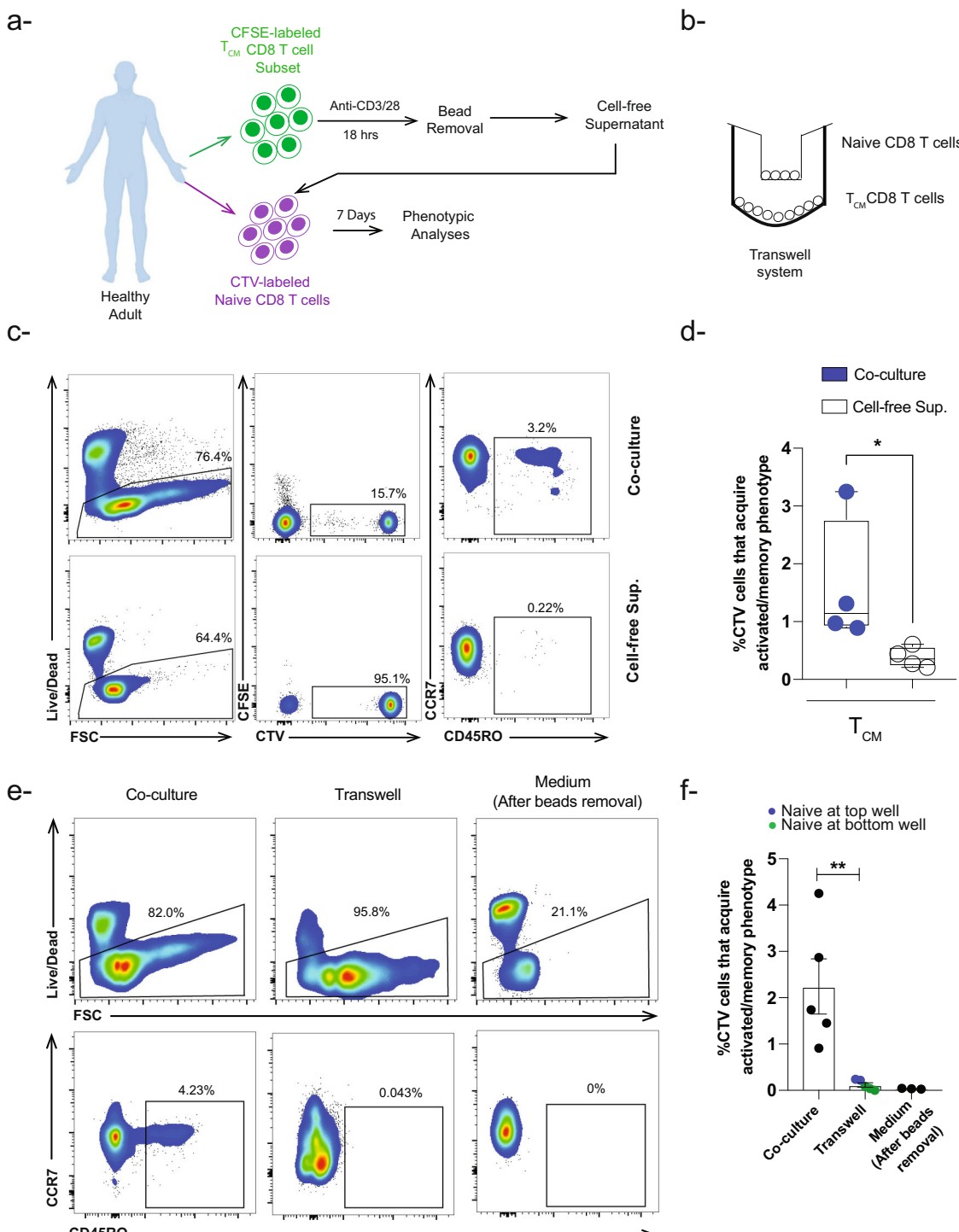

**Fig. 4 Naïve CD8 T cells require cell-to-cell contact to acquire activated/memory phenotype. a** Schematic view of the experimental approach showing that naïve and memory $T_{CM}$ CD8 T cells were sorted from healthy adults then labeled with CTV and CFSE respectively. CFSE-labeled $T_{CM}$ cells were stimulated with anti-CD3/CD28 beads (1:1 ratio). After overnight stimulation, beads were removed by plate magnet followed by collection of cell-free supernatant, which then was added to unmanipulated CTV-labeled naïve CD8 T cells. Following 7 days of incubation, activated/memory T cell phenotype was examined in CTV-labeled naïve CD8 T cells. **b** Schematic view of the transwell system experimental approach showing sorted naïve CD8 T cells were cultured on the top well of transwell system while activated memory $T_{CM}$ CD8 T cells were cultured at the bottom well. **c** Representative flow cytometry plots and **d** Box plot showing the incidence of CTV-labeled naïve CD8 T cells that acquired activated/memory phenotype in the presence of cell-free supernatant from stimulated $T_{CM}$ CD8 T cells compared to co-culture conditions ($n = 4$). **e** Representative flow cytometry plots and **f** Bar-graph showing the incidence of CTV-labeled naïve CD8 T cells that acquired an activated/memory phenotype (upregulation of CD45RO) in the presence of stimulated $T_{CM}$ CD8 T cells but separated from each other using the transwell system compared to co-culture conditions ($n = 5$). Blue closed circles depict conditions were naïve CD8 T cells were cultured at the top ($n = 2$) while green closed circles represent conditions were naïve CD8 T cells cultured at the bottom ($n = 3$). To examine if there was a carry-over effect of beads following beads removal, medium was added to CTV-labeled naïve CD8 T cells after beads removal ($n = 3$). **$P < 0.01$ *$P < 0.05$ Unpaired nonparametric Mann-Whitney test was used. Data are presented as means ± SEM.

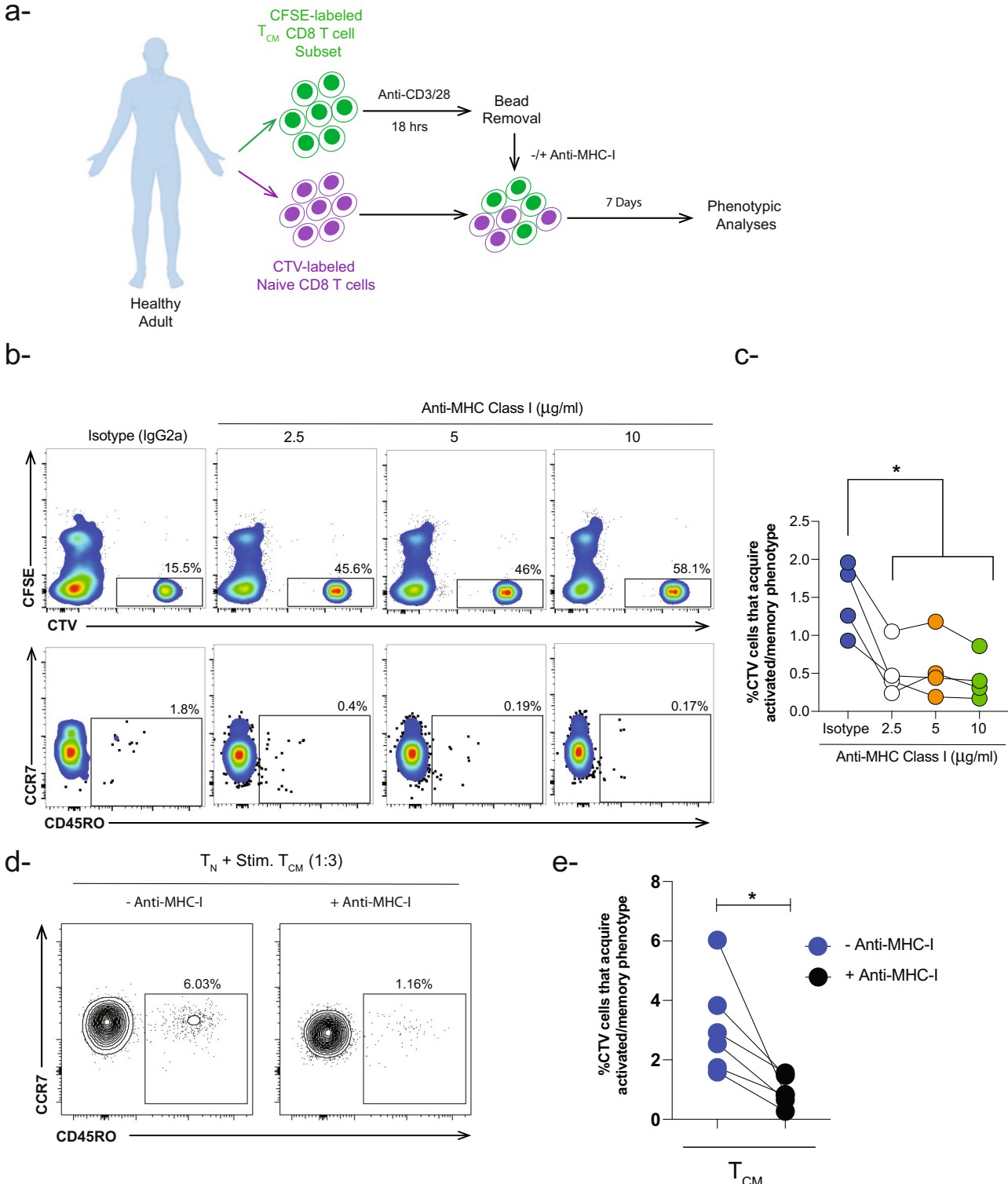

**Fig. 5 Activated/memory phenotype is acquired by naïve CD8 T cells in MHC class I dependent manner. a** Schematic view of the experimental approach shows that naïve and memory $T_{CM}$ CD8 T cells were sorted from healthy adults then labeled with CTV and CFSE respectively. CFSE-labeled $T_{CM}$ cells were stimulated with anti-CD3/CD28 beads (1:1 ratio). After overnight stimulation, beads were removed by plate magnet followed by incubation with CTV-labeled naïve CD8 T cells at ratio 1:1 and 3:1 ratio (memory:naïve) in the presence of different concentrations of anti-MHC class I antibody. Isotype antibody IgG2a was used as a control (2.5 µg/ml). **b** Representative flow cytometry plots and **c** Column graph (individual values) tracking the effect of anti-MHC-I antibody at different concentrations (2.5, 5, 10 µg/ml) on the acquisition of activated/memory phenotype by naïve CD8 T cells compared to isotype control. Ordinary one-way ANOVA Bonferroni multiple comparison test was used *$P < 0.05$ ($n = 4$). **d** Representative flow cytometry contour plots and **e-** Paired analysis showing the incidence of CTV-labeled naïve CD8 T cells that acquired a memory phenotype in the presence of stimulated $T_{CM}$ CD8 T cells (memory:naïve—3:1 ratio) with (black circle) or without (blue circle) anti-MHC Class I antibody 5 µg/ml. Wilcoxon test (paired nonparametric $t$ test) was used *$P < 0.05$ ($n = 6$).

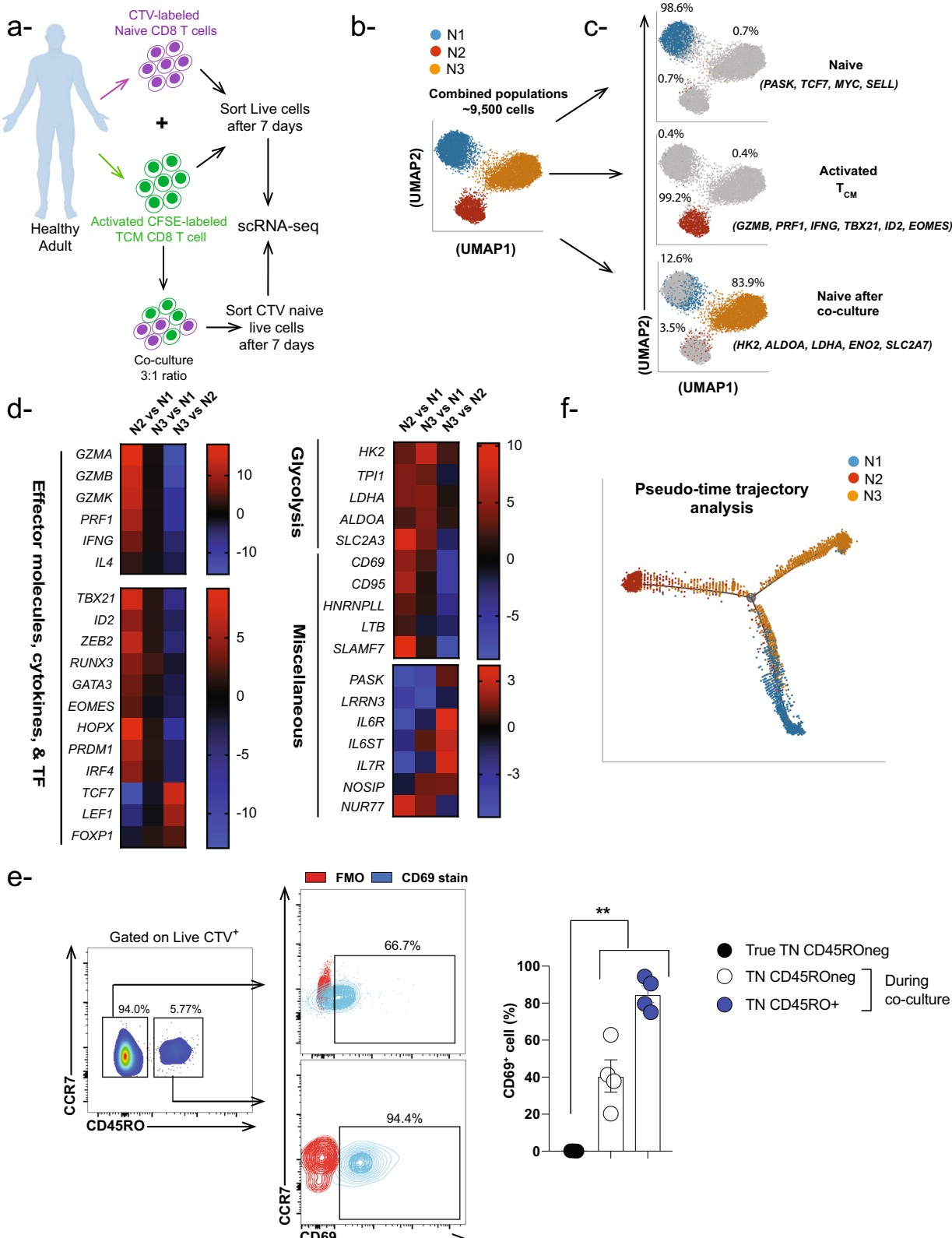

reduction algorithm was applied on sequenced cells from the three samples: Naïve, activated $T_{CM}$, and naïve after co-culture. The resulting UMAP space contained three visually distinct neighborhoods, which were then classified as N1, N2, and N3 (Figs. 6b, S2b). While almost all of the cells from the naïve sample localized in N1 and the activated $T_{CM}$ sample in N2, naïve cells after the co-culture were distributed across the three

neighborhoods (Fig. 6c). Consistent with the cell surface phenotyping (Fig. 1), ~3% of naïve cells after co-culture neighbored activated $T_{CM}$ in N2, but only ~13% neighbored naïve T cells in N1 (Fig. 6c). Instead, the highest percentage of naïve cells after co-culture (~84%) localized to a distinct neighborhood, N3, indicating that they underwent transcriptional changes divergent from those that might have occurred in naïve or activated

**Fig. 6 Activated T$_{CM}$ CD8 T cells induce transcriptional changes in autologous naïve CD8 T cells. a** Experimental setup for scRNA analyses from one healthy donor. **b, c** UMAP analyses (focusing on top 100 genes) showing percentage of each neighborhood per sorted cell population including CTV naïve CD8 T cells cultured alone for 7 days (Blue neighborhood-N1), CFSE activated T$_{CM}$ CD8 T cells cultured alone for 7 days (Red neighborhood-N2), and CTV naïve CD8 T cells sorted from the co-culture mix which contains blue, red and orange (Orange-hybrid) neighborhoods. **d** Heat maps depicting genes upregulated in activated memory vs naive neighborhoods (N2 vs N1), hybrid vs naïve neighborhoods (N3 vs N1), (N2 vs N1) as well as downregulated in hybrid vs activated memory T cells (N3 vs N2). **e** FACS plots and bar-graph showing expression of CD69 cell surface activation marker within CTV$^{+}$ CD45RO$^{neg}$ and CTV$^{+}$ CD45RO$^{+}$ sub-populations following co-culture of CTV-labeled naïve CD8 T cells with CFSE-activated T$_{CM}$ CD8 T cells (3:1 ratio-memory:naïve) compared to naïve CD8 T cells in the absence of activated memory CD8 T cells. Red contour plots represent FMO (full minus one) control, while blue contour plots represent CD69 staining. One-way ANOVA Bonferroni multiple comparison test was used **$P < 0.01$. Data are presented as means ± SEM ($n = 3$-4). **f** Pseudotime temporal analyses showing the differentiation trajectory of N1, N2, and N3 based scRNA-seq gene expression data.

memory cells cultured alone. Thus far, our results imply that the vast majority of naïve CD8 T cells undergo substantial transcriptional changes in the presence of activated memory CD8 T cells, whereby a small proportion becomes transcriptionally similar to activated memory (N2) while the rest take on a unique transcriptional profile (N3).

To confirm the activated/memory transcriptional signature of N2, and to explore the nature of N3, we performed DEG analysis (Suppl. Table 1). Genes differentially expressed in N1, which mostly contained cells from the naïve sample, were associated with the naïve state. These included *PASK, LLRN3, SELL, TCF7,* and *MYC*[15,37]. Compared to N1, N2 exhibited gene upregulation associated with the effector and activated/memory states, such as *GZMB, PRF1, IFNG, TBX21, ID2, EOMES,* and the glycolysis pathway (*HK2, LDHA, ENOA, SLC2A7*), while genes associated with the naïve state were downregulated (Fig. 6d, N2 vs N1). In contrast, when compared to N1, N3 showed upregulation of genes associated with activation (*RUNX3,* and the glycolysis pathway) but not those associated with the effector state despite the well-accepted correlation between glycolysis and T cell effector state (Fig. 6d, N3 vs N1). Moreover, N3 still differed from N2 in that it retained upregulated expression of genes associated with naïve cells including *PASK, TCF7, LEF1,* and *IL7R* (Fig. 6d, N3 vs N2). Of note, we did observe upregulation of the nuclear receptor *NUR77 (NR4A1)* in both N3 and N2 clusters compared to N1 cluster (Fig. 6d, N2 vs N1 & N3 vs N1), which indicates occurrence of TCR-stimulation event[38]. Ingenuity Pathway Analysis (IPA) of N3 identified glycolysis as the top enriched pathway (Fig. S2c), further confirming the activation state of this neighborhood. Therefore, these results demonstrate that a minority of naïve CD8 T cells co-cultured with activated memory CD8 T cells attained a transcriptional profile consistent with the effector or activated/memory state, while the majority acquired a unique transcriptional profile hybrid between naïve and activated memory CD8 T cells.

Since majority of the naïve CD8 T cells that were co-cultured with activated T$_{CM}$ CD8 T cells acquire a unique transcriptional profile i.e., N3 cluster, we thought to validate these changes at a protein level using flow cytometry-based approach. As shown in Fig. 6d, the upregulation of CD69 transcript could be a good candidate to differentiate between N1 (naïve alone) and N3 (unique T cell state). Hence, we repeated the scRNA-Seq co-culture conditions (activated T$_{CM}$:T$_N$-3:1) to examine CD69 cell surface protein expression. Our data demonstrated ~40% of CTV$^{+}$ CD45RO$^{neg}$ cell population expresses CD69. Furthermore, we observed a significant upregulation of CD69 in CTV$^{+}$ CD45RO$^{+}$ and CTV$^{+}$ CD45RO$^{neg}$ sub-populations compared to naïve CD8 T cells in the absence of activated T$_{CM}$ CD8 T cells (Fig. 6e). Additionally, as shown in Fig. 1g, h, CD95 protein expression differentiated between CTV$^{+}$ CD45RO$^{neg}$ and CD45RO$^{+}$ sub-populations. These data suggest that majority CTV$^{+}$ naïve CD8 T cells co-culutred with activated T$_{CM}$ CD8 T cells upregulate CD69 cell surface protein, which is widely

accepted as an activation marker for T cells reflecting ongoing TCR-dependent responses.

To infer the trajectory of cellular differentiation in the three samples and to further confirm that N1, N2, and N3 are distinct neighborhoods, we utilized a pseudotime trajectory inference algorithm (Monocle 2) that is independent of UMAP output. As shown in Fig. 6f, the pseudotime trajectory consisted of three branches and one branching point. Consistent with the UMAP, N1 and N2 cells lie on the ends of two distinct branches in the trajectory, while N3 cells were distributed on the three branches, localizing near the branching point and at the end of the third branch. This trajectory supports the interpretation that the naïve cells co-cultured with activated memory T cells transition into unique and activated/memory T cell states.

## Discussion

Cytotoxic CD8 T cells (CTLs) are classically described as the "serial killers" of the immune system. They play a crucial role in host immune protection against pathogens including viruses, bacteria, parasites, and fungi. Additionally, they can fight tumors if they are not exhausted. Ironically, under certain environmental and genetic conditions they contribute to a wide range of auto-immune diseases e.g., Multiple Sclerosis, Rheumatoid Arthritis, and Type I Diabetes. Furthermore, alloreactive CD8 T cells are considered as one of the main drivers for transplant rejection[4,39-44]. Although, the cytotoxic characteristic features of CTLs are well-defined, their noncytotoxic functions have not been studied extensively. Here, we demonstrated a noncytotoxic role of human activated memory CD8 T cells upon their inter-action with autologous naïve cells where naïve T cells acquire phenotype and transcription profile of two different states: (1) an activated/memory T cell state and (2) a unique transcriptional state that is a hybrid between naïve and effector/activated memory cells.

The concept of noncytotoxic roles of CD8 T cells has been studied previously by other groups as well. These functions can be exerted directly by T cells surprisingly through their killing machinery i.e., effector molecules. For instance, CD8 T cells maintain HSV-1 neuronal latency through degradation of viral proteins in granzyme B (GzmB) dependent-fashion[45]. Further, CTLs can suppress HBV and HCV replication in infected hepa-tocytes in a non-cytopathic manner[46,47]. Alternatively, CTLs can employ their noncytotoxic functions indirectly via cross-talk with other immune and non-immune cells. The Masopust lab and others spearheaded elegant studies showing that tissue-resident memory T cells generate a systemic and/or tissue-resident broad antimicrobial state via recruitment of T and B cells as well as activation of DCs and NK cells[48-52]. Further, memory CD8 T cells can protect dendritic cells (DCs) from cytotoxic T lym-phocytes (CTL)-mediated killing by inducing GzmB inhibitor (Serpin SPI-6/PI-9) expression by the DCs[53,54], while CD8 T cells expressing IL-21 provide help to B cells[55]. Recently, it has been reported that SLAMF7 and IL-6R cell surface expression

distinguishes between cytotoxic and helper CD8 T cells, respectively[56]. Since in our study both naïve and memory CD8 T cells were sorted from the same healthy donors it is likely that upon activation, $T_{CM}$ CD8 T cells developed a noncytotoxic function and presented self-peptides to naïve CD8 T cells acting as antigen-presenting cells (APCs). Our findings not only extend the concept of noncytotoxic functions of CD8 T cells into the context of naïve and memory CD8 T cells, but also provide insights into autoimmune reactivity that could arise after vaccination, infection, or transplantation.

Naïve CD8 T cells are tolerant of self-MHC molecules where their interaction provides low level of signaling that promotes cell survival[57,58]. However, in our in vitro system, naïve CD8 T cells not only survived but also became responsive and acquired activated/memory phenotypic and transcriptomic characteristic features. Indeed, our results are supported by a previous study showing that self-MHC peptide complexes become immunogenic during reduction in the total pool size of mouse T cells, resulting in proliferation and upregulated expression of the CD44 activation marker by CD4 T cells[59]. Furthermore, based on our findings and evidence from the literature, we postulate that the cross-talk between autologous naïve and activated memory T cells could be one mechanism for generation of auto-reactive CD8 T cells where naïve, auto-reactive T cells become activated by autologous activated memory T cells. Our hypothesis is supported by earlier data showing auto-reactive T cells can be readily detected in peripheral blood of healthy individuals[60]. Although our data demonstrate that MHC-I plays a pivotal role in the cross-talk between naïve and activated memory CD8 T cells, we do not overlook the potential minimum contribution of soluble factors, including cytokines in this process, specifically, the frequency of naïve CD8 T cells with activated/memory phenotype was not completely abrogated following MHC-I blocking.

Several pioneer labs including Ahmed, Sprent, Surh, Lefrancois, and others investigated the role of common gamma chain cytokines such as IL-2 and IL-15 and their effect on naïve CD8 T cells phenotype and function[34,61,62]. The common line of evidence that runs through these studies indicates that naïve cells masquerade as memory in the presence of cytokine-rich lymphopenia environment. However, more intriguingly, they showed the necessity of MHC class I in their animal models for this process. For instance, in Murali-Krishna and Ahmed's study, they observed that naïve cells did not undergo T cell activation and LIP upon transfer to irradiated MHC-I knockout (B2m KO) mice, which hints at the importance and necessity of MHC-I even in the presence of cytokine-rich lymphopenic environment[34]. Further, Cho et al recapitulate these observations by transferring naïve CD8 T cells into irradiated MHC-I and Tap-1 Knockout mice in the presence of IL-2 complex (IL-2a/IL-2)[3,61]. These results raise the question of whether there is a link between response to gamma chain cytokines and expression of MHC-I. Indeed, Lefrancois lab elegantly showed that adoptive transfer of naïve CD8 T cells into irradiated B2m KO mice did not proliferate compared to WT mice even following injection with IL-15 complex (IL-15/IL-15Rα)[62]. Thus far, the above-mentioned observations give us a clue toward the role of MHC-I-TCR axis possibly by sensitizing naïve cells to be responsive to gamma chain cytokines. Mathew et al. showed the upregulation of CD122 (IL2/15Rβ) in polyclonal and LCMV gp33-specific CD8 T cells following LCMV acute infection[63]. On the contrary, gamma-chain cytokines such as IL-2 can reduce TCR threshold and sensitize CD8 T cells to be more responsive to low-binding affinity peptides, which could further explain participation of low-affinity self-antigen specific CD8 T cell clones in an autoimmune response[64]. Interestingly our TWS results further suggest that not only MHC-I but also cytokines that require trans-presentation

rather than secretion such as IL-15 could play a role in our phenotype[65–67]. In summary, these studies demonstrate that although cytokines play a role, MHC-I is still necessary.

Another possible scenario that could explain our phenotype is the replenishment of the pool of memory CD8 T cell pool following pathogen control. During homeostasis, naïve CD8 T cells continuously recirculate between the periphery and secondary lymphoid tissue[5,6]. Hence, it is likely they can interact with activated memory CD8 T cells in lymphoid tissue. Following pathogen clearance, large numbers of activated cells die off[68,69]; however, the pool of memory CD8 T cells is required to maintain its numbers. To achieve such homeostasis requires a fine balance between cell death and proliferation[70], which is mediated in part by the common gamma chain cytokines, IL-7 and IL-15[18,71–74]. Our results reveal an additional mechanism besides antigen-independent homeostatic proliferation (Fig. 7).

Broadly, our findings identify a noncytotoxic function of human memory CD8 T cells that could represent one mechanism whereby memory cells are 'conscious' to set the activation threshold of autologous naïve cells. This could be relevant to several scenarios beyond homeostasis settings; for instance, our results could also explain the activation of autoreactive T cells and the surprising emergence of autoimmune diseases following vaccination[75]. Additionally, the recurrence of autoimmune diseases following organ transplantation[76–80] could be explained by the cross-talk between donor-specific alloreactive memory T cells and auto-reactive naïve T cells. Finally, these results lay the foundation and open future avenues to uncover additional cross-talks between activated memory T cells and a wide spectrum of other immune and non-immune cell types.

## Materials and methods

**Isolation and flow cytometric analysis of human naïve and memory CD8 T cell subsets from healthy adult blood.** This study was approved by the Institutional Review Board (IRB#00608014) of The University of Pittsburgh, School of Medicine. Consent was obtained from all donors. Human mononuclear cells were isolated from peripheral blood of healthy adults followed by CD8 T cell purification using Mojo Human total CD8 T cell negative selection enrichment kit (BIOLEGEND). After enrichment of CD8 T cells, naive and memory CD8 T cell subsets were FACS-purified using fluorochrome-conjugated antibodies against the following cell surface markers, as previously described[14,15]. CD8 $T_N$ cells were phenotyped as live CD3+, CD8+, CCR7+, CD45RO−, and CD95− cells; effector memory ($T_{EM}$) CD8 T cells were phenotyped as live CD3+ CD8+, CCR7-, and CD45RO+ cells; central memory ($T_{CM}$) cells were phenotyped as live CD3+, CD8+, CCR7+, and CD45RO+ cells; whereas stem cell memory ($T_{SCM}$) cells were phenotyped as live CD3+, CD8+, CCR7+, CD45RO-, and CD95+ cells. Sorted cells were checked for purity (i.e., samples were considered pure if >95% of the cells had the desired phenotype).

**Cell proliferation dye labeling and generation of polyclonal activated memory CD8 T cells.** Sorted naïve and memory CD8 T cell subsets were labeled with cell trace violet (CTV) dye at 2.5 µM and carboxyfluorescein succinimidyl ester (CFSE) dye at 1 µM respectively (Life Technologies). To generate polyclonal activated memory CD8 T cells, each memory CFSE-labeled subset was stimulated in vitro using Dynabeads Human T-Activator anti-CD3/CD28 coated magnetic beads [1:1 ratio] (Gibco) in a 96-well round-bottom plate for 18 h, followed by bead removal using a plate magnet (Thermo-Fisher Scientific).

**Co-culture of activated memory CD8 T cells with naïve CD8 T cells, flow cytometry, and image stream analysis.** Activated memory CD8 T cells were co-cultured with resting naïve CD8 T cells at 1:1 and 3:1 ratios (Memory:Naïve). The cells were maintained in complete RPMI-1640 culture (cRPMI) medium containing 10% FBS (Atlanta Biologicals), penicillin-streptomycin, and gentamycin (Lonza) at 37 °C and 5% CO2 for 7 days. To examine naïve CD8 T cell phenotype by flow cytometry (BD Fortessa), Live/dead staining was performed (Ghost dye Violet 510, TONBO) and cell surface staining was conducted using the following fluorochrome-conjugated antibodies: CCR7-PE (Biolegend-Clone: G043H7), CD45RO-APC (Biolegend-Clone: UCHL1), and CD95-PECy7 (Biolegend-Clone:DX2). In some experiments, CD69 PE-Cy7 (Biolegend-Clone: FN-50) was used. In addition, for image stream analysis, an aliquot of stained cells was acquired in an Amnis Mark II image stream analyzer and examined with Amnis IDEAS software (Luminex).

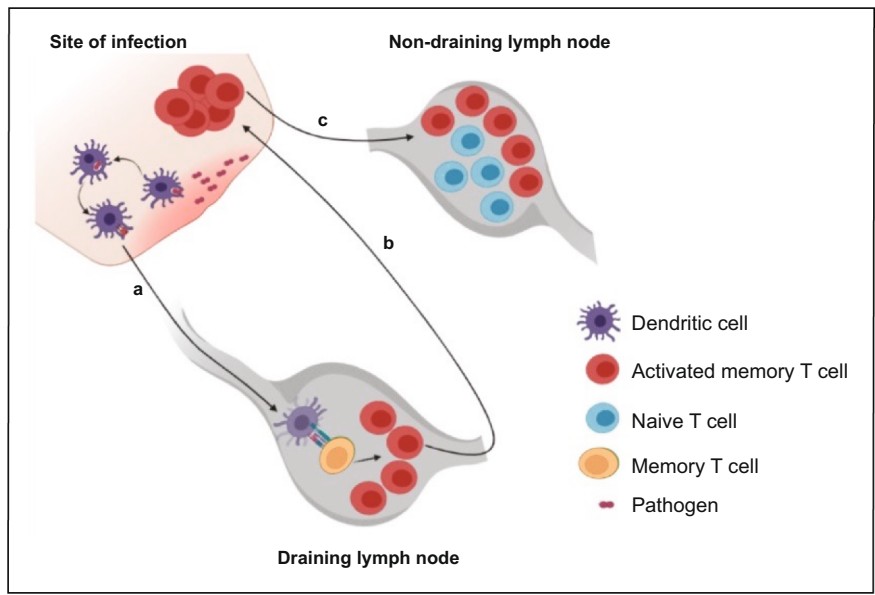

**Fig. 7 Proposed model postulating the cross-talk between naïve and activated memory CD8 T cells. a** Upon pathogen re-exposure memory CD8 T cells become activated in the draining lymph nodes and **b** migrate to the site of infection to control the pathogen. **c** A proportion of the activated memory CD8 T cells migrates to antigen-free lymph nodes where they interact with naïve CD8 T cells resulting in acquisition of an activated/memory phenotypic properties by naïve CD8 T cells. The figure was created by BioRender under University of Pittsburgh license.

**Blocking proliferation of $T_{CM}$ CD8 T cells**. We blocked the proliferation of $T_{CM}$ CD8 T cells using three approaches: (**1**) $T_{CM}$ CD8 cells were resuspended as 150–300 K in 500 µl cRPMI medium followed by gamma irradiation at 2000 rads. Cells were then washed once with 14 ml cRPMI and counted for activation with anti-CD3/CD28 beads in a 96-well round-bottom plate. After 18 h, beads were removed by plate magnet then the cells were co-cultured with naïve CD8 T cells at 3:1 ratios (Memory:Naïve). Non-irradiated cells were used as controls. (**2**) CTV-labeled naïve CD8 T cells were co-cultured with activated $T_{CM}$ CD8 T cells (1:1 ratio) in the absence or presence of different concentrations of CsA 10 ng/ml and 100 ng/ml (Sigma-Aldrich). These concentrations were selected based on previous studies showing potent inhibition of T cell proliferation[81,82]. (**3**) Anti-human IL-2 antibody (R&D-clone#5334) was added to the co-cultures to neutralize IL-2. According to manufacturer's recommendations 0.01–0.03 µg/ml antibody is sufficient to block 2 ng/ml of IL-2. In our experimental setup, we thought to add 5X (0.1 µg/ml) more to ensure complete neutralization of IL-2. IgG2a Isotype controls were used at the same concentration.

**Measurement of IL-2 in the supernatant of activated $T_{CM}$ CD8 T cells**. IL-2 levels were measured with BD Cytometric Bead Array (CBA) kit. Briefly, 50 K $T_{CM}$ CD8 T cells were stimulated with anti-CD3/CD28 beads in a 96-well round-bottom plate. After 18 h, beads were removed by plate magnet then cultured alone for 7 days then supernatants were collected and stored at −80 Celsius. At the day of the experiment, 50 µL of thawed supernatant was added to 50 µl of IL-2 capture beads (beads coated with anti-IL-2 antibody) then incubated for 2 h at room temperature in dark. Subsequently, 50 µL of phycoerythrin (PE) detection reagent was added then incubated for 1 h at room temperature in the dark. The samples were washed with 1 mL of wash buffer and centrifuged for 5 min at 200 *g*. After discarding the supernatant, the bead pellet was resuspended in 100 µL buffer. Measurements were performed on the Flow Cytometer (BD Fortessa) and then analyzed by FCAP Array™ Software (BD Bioscience). IL-2 concentration was measured by its fluorescent intensity. Calibrated standard curve was generated using the IL-2 standard serial dilutions.

**Addition of rhIL-2 to the co-cultures**. Briefly, CTV-labeled naïve CD8 T cells were co-cultured with unstimulated $T_{CM}$ CD8 T cells (1:1 ratio) then recombinant human IL-2 (Peprotech-200-02) was added at final concentration 1 ng/ml and 10 ng/ml. Following 6 days of co-culture, the naïve CD8 T cells were phenotyped using anti-CCR7 and CD45RO fluorochrome-conjugated antibodies.

**Generation of cell-free supernatant and Trans-well system**. Cell-free supernatant was obtained from activated $T_{CM}$ CD8 T cells. Briefly, two wells of $T_{CM}$ CD8 T cells (each at 50 K cells/200 µl) were activated as described above. Supernatant (total of 300 µl) was pooled from both wells followed by two rounds of centrifugation at 450 × *g* to avoid cell contamination. One-third of the volume (100 µl) was used and added to CTV-labeled naïve CD8 T cells. Regarding the trans-well system, CTV-labeled naïve CD8 T cells were cultured in the top chamber of the

Trans-well system (3 µm pore size, Corning) while $T_{CM}$ CD8 T cells were activated in vitro as described above while activated $T_{CM}$ CD8 T cells were cultured in the bottom chamber. In other experiments, naïve cells were cultured in the bottom while memory at the top. Fresh culture media was added every other day to the top chamber to avoid media evaporation. Following 7 days of culture, the naïve CD8 T cells were phenotyped using anti-CCR7 and CD45RO fluorochrome-conjugated antibodies.

**Blocking MHC class I**. Briefly, CFSE-labeled activated $T_{CM}$ CD8 T cells were co-cultured with CTV-labeled naïve CD8 T cells (1:1 ratio) in the presence or absence of anti-MHC class I antibody at a final concentration 2.5, 5, 10 µg/ml (Ultra-LEAF clone W6/32-Biolegend) at the beginning of the 7 days co-culture period. As controls, isotype IgG2a (Biolegend) was added at 2.5 µg/ml final concentration. Separate experiments were done at (memory:naïve—3:1 ratio) in the presence or absence of anti-MHC class I antibody at a final concentration 5 µg/ml.

**Cell preparation and sorting for scRNAseq**. CD8 T cells subsets were sorted from one healthy donor. Activated $T_{CM}$ cells (150 K) were labeled with CFSE and co-cultured with resting CTV-labeled naïve CD8 T cells (50 K) at a 3:1 ratio respectively for 7 days. As for controls, isolated naïve T cells or activated $T_{CM}$ were cultured separately in similar conditions. Thereafter, live naïve and activated $T_{CM}$ cells were sorted, while only CTV$^+$ cells (which included both CTV$^+$ CD45RO$^-$ and CD45RO$^+$ subpopulations) were sorted following co-culture of naïve with activated $T_{CM}$ cells. Each sorted sample was then labeled with cell hashing antibodies (Totalseq-C, Biolegend) for later demultiplexing, then pooled together and sequenced in one flow cell.

**scRNAseq and data preprocessing**. scRNAseq was performed using 10× Genomics Single Cell 5' solution, version 1, according to the manufacturer's instructions with 4000 sorted cells loaded for naïve or activated $T_{CM}$ and 9000 cells for naïve after co-culture. mRNA and hashtag oligos cDNA libraries from pooled samples were sequenced on the NovaSeq6000 Platform (Illumina). Single-cell raw matrix files were obtained using the Cell Ranger's pipeline with alignment to the human reference hg38. The raw matrix was then preprocessed and analyzed using Partek Flow v10.0.21.0801 (Partek). Quality control was done using knee point and EmptyDrops[83] to exclude empty droplets. After centered log-ratio (CLR) normalization of the cell hashing count matrix, Hashtag demultiplexing was performed by using an implementation of the algorithm used in ref.[84]. Multiplets were excluded based on the cell hashing classification and applying an inclusion filter on counts per cell (600–15000) and detected genes per cell (500–4000). Cells with greater than 10% mitochondrial gene expression were excluded to eliminate dead or apoptotic cells. The resulted gene expression matrix was of 9455 cells by 19,327 genes. Single-cell gene expression counts were normalized by $\log_2$ (counts per million + 1).

**Bioinformatic analyses**. Dimensionality reduction was performed using Uniform Manifold Approximation and Projection (UMAP) on the top 100 and 2000 genes of highest variance utilizing the following parameters: local neighborhood size = 15, minimal distance = 0.1, distance metric = Euclidean, initialization = random. By visualizing cells in two dimensions based on their gene expression, UMAP allowed for the detection and classification of visually distinct cell neighborhoods. To compare the gene expression of each neighborhood, we performed Partek Flow's Gene Specific Analysis. The resulting fold changes were represented using heatmaps that were generated in Graphpad Prism (Version 8.3). Canonical pathway activity was investigated using Ingenuity Pathway Analysis (Qiagen). Pseudotime trajectory inference (Monocle 2) was utilized to demonstrate the cellular differentiation states and trajectory of differentiation in our sequenced samples.

**Statistics and reproducibility**. All statistical analyses were performed using GraphPad Prism software (Version 8.3). Unpaired nonparametric Mann-Whitney and paired nonparametric Wilcoxon tests were used for two groups comparison. For comparisons of more than two groups, ordinary one-way ANOVA Bonferroni multiple comparison test was used. Data were presented as means ± SEM. $*P < 0.05$, $**P < 0.01$, and $****P < 0.0001$. In flow cytometry experiments, at least four biological replicas were used to ensure reproducibility within a given experiment. In regards to the scRNA-Seq experiment, gene expression changes across the neighborhoods were validated at a protein level using flow cytometry-based approach.

**Reporting summary**. Further information on research design is available in the Nature Research Reporting Summary linked to this article.

## Data availability

Single cell RNA sequencing raw data generated in this study has been deposited to the NCBI Gene Expression Omnibus (GEO) accession number GSE202548. All data used in graphs are provided in Supplementary Data 1.

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

## Acknowledgements

We would like to thank Dr. Noor A. Abdelsamad (Bayer Crop Science, CA, USA), Drs. Adrian E. Morelli and Martin H. Oberbarnscheidt (Starzl Transplantation Institute, University of Pittsburgh, School of Medicine), Dr. Hazem Ghoneim (Ohio State University), Dr. Ali Ellebedy (Washington University) and Dr. Ben Youngblood (St. Jude Children's Research Hospital) for fruitful discussions and suggestions over the manuscript. For flow cytometry technical assistance, we would like to thank Dewayne Falkner and Tim Sturgeon for flow sorting and Aarika McIntyre for image stream analyses (Unified Flow Cytometry Core Facility, University of Pittsburgh, School of Medicine). For the scRNA-Seq library preparation and sequencing, we thank the Rheumatology single cell core (Tracy Tabib) and UPMC genomics core. This work is supported by Community Liver Alliance [CLA] Foundation, The American Gastroenterological Association (AGA), Pittsburgh Liver Research Center (PLRC) Pilot and Feasibility Grant, University of Pittsburgh, School of Medicine (PI: Hossam A. Abdelsamed, NIH/NIDDK P30-DK1120531). The work also benefited from IMAGE STREAM MARK II funded by NIH 1S100D019942-01 (PI: L. Borghesi).

## Author contributions

K.S. and M.A. performed the experiments, interpreted the results, and helped write the manuscript. A.F.Z. and M.J. helped in performing the experiments. C.M. collected and analyzed advanced image stream flow cytometry and beads array data. K.I.A-D. guided, led, and performed the bioinformatic analysis of the scRNAseq data and helped in writing scRNA-seq section. S.L. and A.H.A. helped in the bioinformatic analyses. D.M.M., A.W.T., F.G.L., and H.A.A. supervised the study, interpreted the results, and helped in writing the manuscript. H.A.A. conceived, designed, and supervised the project, interpreted the results, and wrote the manuscript.

## Competing interests

The authors declare no competing interests.
