## [Peer Review File · Communications Biology]

Reviewer #2 (Remarks to the Author):

A novel non-cytotoxic function of human CD8 T cells

Brief summary and overall impression of the work

This is a well-conducted and well-written piece of original research that examines the question of how activated CD8 memory cells exert control over naïve CD8 T cells. The authors demonstrate that activated CD8 memory cells control activation of naïve CD8 T cells primarily through an MHCI-dependent mechanism. Upon interaction a minority of the naïve cells acquire an activated/memory phenotype, another minor population is still transitioning to its final phenotype(s) at the timepoint analysed and a majority develop into a transcriptionally distinct subset.

This research will be of interest to many immunologists as these interactions between naïve and activated memory cells would be expected to occur in vivo in a variety of situations. Statistics are appropriate throughout. The level of detail provided is sufficient for other researchers to reproduce the work, once minor comments are addressed.

Specific comments

Major points

1. Line 29, 30, 62, 265, 272, 280 and others-I do not feel that the data support the idea that 83% are intermediate between naïve and activated memory, as some of these cells have the appearance of taking an entirely different transcriptional path (and would not end up occupying the activated memory space). In other words: I would agree from the UMAP plots and fig4F it appears that the majority of "naïve after co-culture" cells in N3 have started from N1 (were once naïve). However, whilst a minority of these cells appear to be heading towards becoming N2 (i.e. activated TCM), the majority are heading in a different developmental pathway. This is essentially what the authors say in lines 245-250, but the narrative is then confused by the discussion of intermediate cell types that lie between naïve and activated memory. Please could the authors rephrase and discuss where they think the final destination of the majority of subset N3 subset is likely to be?

2. scRNAseq: It is not clear in the methods or results how many donors were used and if they were pooled together without hashing (i.e. if the hashing step refers to the different cell types).

- If this represents at least 3 individual donors, with similar percentages in each cell subset, then please just clarify this.
- If it represents one or two donors- then I would not reasonably expect this to be repeated using scRNAseq due to the high costs involved. However, the authors should pick some key markers that represent each of the three neighbourhoods and confirm the results using flow cytometry of at least three individual donors.

Minor points

1. line 1: the title could be more descriptive of the function

2. line 57 please could the authors clarify why they have focused on CD8+ cells rather than CD4+ cells

3. line 69- please clarify that informed consent was obtained from all donors

4. line 105 please include cell densities as well as cell number throughout and on this line explain the volume used

5. line 114- please state the manufacturer or clone id.

6. line 117- please state the number of independent replicates and number of donors used (see also major point above)

7. line 130-please state average number of reads/cell

8. line 133- please explain in more detail the default settings on partek flow, the approach to quality control, demultiplexing and how multiplets were identified.
9. line 140 please explain why UMAP was performed on the top 100 genes of highest variance? In Seurat one would generally use the top 2000 genes for downstream analysis, but perhaps the methodology differs for Partek flow?
10. line 186, 292 This sentence reads somewhat as if these cells lack cytotoxic capability, whereas actually this is an additional function to control naïve cells, independent of the cytotoxic function. Please could these sentences be clarified?
11. line 226 please state the values in the text
12. line 312 in addition to MHCI interaction and minor contributions from soluble factors, might other cell contact dependent interactions play a minor role in the cross talk between naïve and activated memory cells? Please discuss.
13. Throughout the figures- I do not feel it is helpful to put the fold change above the asterisks
14. Figure 2B- the CFSE graphs should be described more fully in the text
15. Figure 2B/D- I appreciate that this CFSE/CTV experiment had to run for 7 days and therefore the divisions are not as sharp as would be seen at an earlier timepoint. However, some staining looks very blurry/low. Please could a supplementary figure be provided that confirms uniform staining at day 0 and/or well-defined division around day 3?
16. Figure 4A- is confusing, some additional arrows would clarify where the bottom pool of cells come from?
17. Figure 4B/C further on the point about only using 100 genes for the UMAP projection (and without knowing the default setting for Partek flow), these UMAP plots appear to be somewhat underclustered? Altering the analysis parameters may allow more intermediate populations to be revealed (and possibly give the option of visualising the trajectory analysis on the UMAP plot)?

Stephanie J Hanna

Reviewer #3 (Remarks to the Author):

Review:

This is an interesting and well-presented study which highlights the potential for human memory CD8+ T cells to directly activate their naïve counterparts, driving them towards a memory phenotype via autologous MHC class I interactions and in the absence of a conventional APC priming event. The series of in vitro experiments have been well-designed and clearly described and includes imagestream and transcriptomic analysis of the activated naïve T cells. The paper does have the potential to add to our understanding of T cell biology, however, I do have some comments that the authors would need to address.

General comments:

1. Although it is clear that cell contact is required to initiate the activation of autologous naïve T cells in co-culture, the potential downstream role of IL2 in driving naïve CD8 T cells towards a memory phenotype, which has already been described in the literature, should be acknowledged/discussed.
2. The contact-dependent mechanism has been attributed here to autologous MHC-Class I/TCR interaction between the activated CD8+ memory and naïve CD8+ T cells but there is no clear explanation of why this does not occur in the absence of prior CD8 memory cell activation (non-activated TCM will still express MHC class I). Can the authors comment on this? Presumably there are additional factors activated by the polyclonal stimulation of memory cells prior to co-culture?

Can the effect also be blocked with anti-IL2 or by blocking costimulatory receptor signalling?
3. The authors claim, based on transcriptomic data, that this is a potential non-cytotoxic role for CD8+ T cells but lack of cytotoxicity has not been formally demonstrated at the protein or functional level.

4. In general there is a need to be clear in the figure legends about the number of biological replicates/samples that data represent.

Minor comments:

Figure 1A

If plate magnet is used to remove beads, is the subsequent co-culture performed in the same media? If so, how can the authors control for contribution of IL2 or other cytokines from the stimulated memory Teff population?

Figure 1B

The authors correctly do not gate CTV- naïve T cells to avoid including activated/proliferated memory CD8 T cells which might overlap with the CTV- population. However, this would not control for auto fluorescence of the activated/proliferated memory T cells in the CTV channel. An additional control of stim CFSE labelled memory cells alone (with no TN) would control for this.

Figure 1D

The imagestream data is an excellent addition and helpful in demonstrating a clear surface upregulation of CD45RO on a CTV+ (thus previously naïve) single cell. However, what is this figure representative of (i.e. how many biological/technical replicates)?

Figure 2

FigD needs further clarification of gating since it is difficult in this bottom left quadrant to be completely clear what is divided memory CFSE+ (now CFSE-) cells versus proliferated naïve (CTV+ now CTV-) cells and it is not clear how the % CD45RO+CTV+ Naïve T cells in Fig 3E have been derived because the gating of naïve/CTV+ T cells is not shown. Additional timepoints should be considered here - gating of CFSE dividing cells would be much clearer for example at day 3-5 when memory cells are divided but not completely CFSE negative.

Figure 3

Fig 3D Can the authors comment on the direction of blocking since presumably pan-class I antibody could potentially block MHC Class I on both the naïve and memory T cells. As per my earlier comment, since unstimulated memory CD8 T cells also express MHC Class I can the authors comment on the need for stimulation of the memory cells in inducing naïve T cell activation?

Figure 4

If 85% of the naïve T cells undergo a transcriptomic shift to N3 can some of these changes be validated at the protein level? For example, upregulation of CD69 would be easy to demonstrate. Lactate can be used as a readout of glycolysis. As per my earlier comment the authors should be clear that non-cytotoxic function has not been demonstrated functionally but rather transcriptomically.

Reviewer #1 (Remarks to the Author):

This is a simple, yet interesting study, showing that CD8 T cells with a naive phenotype undergo phenotypic and transcriptional changes upon co-culture with autologous activated memory CD8 T cells, thus describing a novel, non-cytotoxic function of CD8 T cells. Overall, the manuscript is well written, the results are new and may prove useful to understand cellular communications in protective immune responses and autoimmunity.

We are very thankful for the positive evaluation and insightful comments posed by the reviewer, which actually enhance the manuscript significantly in the right direction. Further, we are glad that the reviewer found our study novel, interesting, and well-written. Please kindly find below point-by-point response to your suggestions/comments.

This notwithstanding, there are points authors need to address to improve this manuscript. Specific points follow.

1) Authors should explain why UMAP analysis was performed on the top 100 genes.

1) We appreciate the reviewer's question, which actually gives us the opportunity to discuss our rationale behind using this kind of computational analyses. Given the fact that the scRNAseq dataset is of highly similar T cells, focusing on the top 100 genes by variance restricts the projection of the T cell neighborhoods to the most variable genes. In other words, our approach focuses the algorithm on the most prominent transcriptomic changes in the dataset, which consequently reduces the chance of projecting a neighborhood based on false or minimal transcriptomic changes. Furthermore, restricting the algorithm to the top 100 variable genes reduces its computational cost and increases its speed of execution.

*Nevertheless, to validate the projection by UMAP included in the manuscript we ran a UMAP on the top 2000 genes using the Seurat pipeline specifications as recommended by reviewer #2 (Minor comments point#17). We also performed a PCA on all the genes in the dataset. We color highlighted the three neighborhoods (N1, N2, N3) as classified in the UMAP done on the top 100 genes (which is included in the manuscript) on both the PCA on all the genes and UMAP of the top 2000 genes by variance shown below (**Figure 1**). It is apparent that the projection and neighborhood classification is consistent regardless of number of genes utilized in the UMAP. Using PCA, we can clearly see that even when we use a linear dimensionality reduction technique these three different states are clearly visible as well. To note we utilized different hyperparameters as recommended by reviewer #2 and obtained similar results (data not shown).*

2) How many donors were the cells for scRNA seq from.

2) We used one donor because of the high expense of the experiment. We have indicated that scRNA-Seq was performed on one donor in the revised methods section as well as figure legends. Further, to circumvent the high cost of repeating the experiment, we selected DEGs such as CD69 and CD95 and confirm their protein expression by flow cytometry. Please kindly refer to Reviewer#2 major point#2 as well as Reviewer#3 Fig.4.

3) How do the three branches of the pseudotimed trajectory fit with the pseudotimed differentiation trajectories of CD8 T cells from the peripheral blood of healthy donors (relevant data can be extracted from several scRNAseq datasets of healthy donor's PBMC).

3) We thank the reviewer for such valuable suggestion to draw parallels between our dataset and publicly available datasets. Unfortunately, there are no available datasets for sorted CD8 T cells from PBMCs of healthy human donors. However, there are datasets of PBMCs from healthy donors that are not sorted, these contain a small number of T cells as compared to our current study. The combination of these datasets will not result in capturing a good representation of the CD8 T cells in peripheral blood in terms of sampling and transcriptomic data information quantity and quality. In addition, these datasets were generated on different platforms resulting in significant batch effects. Correcting for these batch effects will introduce errors. None of the datasets found are from the sequencing platform we used in our study (Novaseq6000).

However, we downloaded three PBMCs datasets from healthy donors generated on the Nextseq 500 (from Wang et al, Nat Comm 2021, accession number GSE168732) [1]. In those experiments, a similar number of PBMCs from individual donors were sequenced in different batches. Initial analysis showed significant batch effects. We employed batch effect correction (Seurat3 Integration) and focused on the CD8a and CD8b expressing neighborhoods of cells that also lack of expression of MHCII related genes. Only two CD8 expressing neighborhoods were isolated from the PBMCs (**Figure 2**). One neighborhood was found to be of naïve CD8 T cells and one that is of circulating memory CD8 T cells (distinction between naïve and memory T cells was based on TCF7 and LAG3 expression respectively). In total there were 3107 CD8 T cells of which ~1000 are memory and ~2000 are naïve. Trajectory inference using monocle 2 on the CD8 T cells from healthy donors reveals 3 states and one branching point. The memory CD8 T cell neighborhood is mostly in one state and the naïve CD8 T cell neighborhood is split between two states (see trajectory below). Although this is consistent with what we saw in our dataset, we cannot confidently interpret these results given the reasons mentioned in the previous paragraph.

4) In the absence of a possible underlying mechanism, caution should be taken on the possible correlation between the capacity of memory cells to induce a memory phenotype in T naive cells and their proliferative capacity. Although this correlation is statistically significant, it is not such impressive ($R^2 = 0.36$).

4) We are very appreciative for the reviewer's constructive comments and suggestions. Indeed, we addressed this important point in the revised manuscript by performing new set of experiments to examine a cause-and-effect relationship between proliferation of T_{CM} CD8 T cells and their capacity to change the phenotype of naïve CD8 T cells. In the course of these experiments, we blocked the proliferation of T_{CM} CD8 T cells using three different approaches.

*In the first approach, we irradiated activated T_{CM} CD8 T cells and co-culture them with CTV-labeled naïve CD8 T cells. As controls, we used non-irradiated activated T_{CM} CD8 T cells. As shown in **fig. 3A-B** (also added to the manuscript as figure 3A-B), the frequency of naïve CD8 T cells that acquired activated/memory phenotype dramatically decreased in the presence of activated irradiated (non-proliferating) T_{CM} CD8 T cells compared to activated non-irradiated (proliferating) T_{CM} CD8 T cells.*

*Secondly, we validated our results using the immunosuppressive drug Cyclosporin A (CsA), a calcineurin inhibitor that suppresses T cell proliferation [2, 3]. In this experimental setup, we co-cultured naïve CD8 T cells with activated CFSE-labeled T_{CM} CD8 T cells in the absence or presence CsA (100 ng/ml). In the presence of CsA, we observed a significant decrease in the proliferation of activated memory T_{CM} CD8 T cells as well as naïve CD8 T cells with acquired activated/memory phenotype compared to co-culture conditions without the drug (**Fig. 3C-D**). (Also added to the manuscript as Suppl. figure 3A-B).*

*Since gamma chain cytokines such as IL-2 plays an important role in survival and proliferation of T cells [4, 5], while CsA abrogates T cell proliferation through suppression of IL-2 production and other cytokines [6-9], we took a third approach and examine the effect of IL-2 blockade on our phenotype during the co-culture. To achieve such aim, we activated CFSE-labeled T_{CM} CD8 T cells then co-culture them with CTV-labeled naïve CD8 T cells in the presence of anti-human IL-2. As controls, we used isotype antibody at the same concentration. Following 6 days of the co-culture, we examined the proliferation capacity of CFSE-labeled T_{CM} CD8 T cells as well as the phenotype of CTV-labeled naïve CD8 T cells. We observed a significant decrease in the proliferation capacity of memory T cells and concurrently the frequency of naïve CD8 T cells with an activated/memory phenotype was much less compared to isotype controls. These results demonstrate that IL-2 plays an indirect role in the acquisition of activated/memory phenotype by naïve CD8 T cells via proliferation of memory CD8 T cells (**Fig. 3E-F**). (Also added to the manuscript as Suppl. figure 3C-D).*

The following paragraphs were added to the revised manuscript lines 272-315:

The direct correlation relationship between the proliferation capacity of memory CD8 T cells and the naïve CD8 T cell phenotype prompted us to ask what will happen if we block the proliferation of activated memory CD8 T cells, could this impair their capacity in influencing the phenotype of naïve CD8 T cells. To answer this question, we irradiated activated CFSE-labeled T_{CM} CD8 T cells. As controls, naïve CD8 T cells were co-cultured with activated non-irradiated T_{CM} CD8 T cells. Indeed, upon irradiation of T_{CM} CD8 T cells, we observed a dramatic reduction in the frequency of naïve with activated/memory phenotype compared to non-irradiated controls (Fig. 3A-B**).**

To further validate our findings, we used the immunosuppressive drug Cyclosporin A (CsA), a calcineurin inhibitor that suppresses T cell proliferation [2, 3]. In this experimental setup, we co-cultured naïve CD8 T cells with activated CFSE-labeled T_{CM} CD8 T cells in the absence or presence of different concentrations of CsA (10 ng/ml and 100 ng/ml). In the presence of CsA, we observed a significant decrease in the proliferation of activated memory T_{CM} CD8 T cells as well as naïve CD8 T cells with acquired activated/memory phenotype compared to co-culture conditions without the drug (**Suppl. Fig. 3A-B**). These results demonstrated that the proliferation capacity of activated memory T_{CM} CD8 T cells plays an important role in acquisition of activated/memory phenotype by naïve CD8 T cells.

Since gamma chain cytokines such as IL-2 plays an important role in survival and proliferation of T cells [4, 5], while CsA abrogates T cell proliferation through suppression of IL-2 production and other cytokines [6-9], we thought to examine the effect of IL-2 blockade on our phenotype during the co-culture. To achieve such aim, we first determined the levels of IL-2 in the supernatant of activated T_{CM} CD8 T cells following beads stimulation as a guidance for how much anti-IL-2 we should add during the co-culture conditions. As expected, we observed a significant increase in the levels of IL-2 in the supernatant of stimulated T_{CM} CD8 T cells with an average of 1300 pg/ml compared to unstimulated controls (**Suppl. Fig. 3C**).

We next activated CFSE-labeled T_{CM} CD8 T cells then co-culture them with CTV-labeled naïve CD8 T cells in the presence of anti-human IL-2. As controls, we used isotype antibody at the same concentration. Following 6 days of the co-culture, we examined the proliferation capacity of CFSE-labeled T_{CM} CD8 T cells as well as the phenotype of CTV-labeled naïve CD8 T cells. We observed a significant decrease in the proliferation capacity of memory T cells and concurrently the frequency of naïve CD8 T cells with an activated/memory phenotype was much less compared to isotype controls. These results demonstrate that IL-2 plays an indirect role in the acquisition of activated/memory phenotype by naïve CD8 T cells via proliferation of memory CD8 T cells (**Fig. 3C-D**).

However, our approach did not address whether blocking IL-2 will have a direct effect on naïve CD8 T cells to acquire activated/memory phenotype independent of the proliferation of memory CD8 T cells. Hence, we added IL-2 cytokine (1ng/ml and 10ng/ml) to the co-culture conditions of naïve with unstimulated T_{CM} CD8 T cells for 6 days. In this experimental setup, we neither observe an increase in the frequency of naïve CD8 T cells with acquired activated/memory phenotype nor proliferation of T_{CM} CD8 T at both concentrations compared to regular co-culture conditions (**Fig. 3D-E**). These results suggest that MHC-TCR axis (signal 1) and probably other soluble factor(s) could be the early events required to initiate the proliferation of T_{CM} CD8 T cells and hence acquisition of activated/memory phenotype by naïve CD8 T cells. Thus far, our data reveal a cause-and-effect relationship between acquisition of activated/memory phenotype by naïve CD8 T cells and the proliferation capacity of activated memory T_{CM} CD8 T cells.

Fig.3: Representative FACS plots, paired analyses, and bar-graphs showing the effect of blocking T_{CM} CD8 T cell proliferation on acquisition of activated/memory phenotype by naïve CD8 T cells using three different approaches (**A-B**) irradiation of T_{CM} CD8 T cells, (**C-D**) Cyclosporin treatment, and (**E-F**) IL-2 neutralization.

4) In addition, since authors have not addressed (although they do not exclude, see page 15) the participation of cytokines secreted by activated memory T cells, this part should be moderated.

We completely agree with reviewer that we should not overlook the role of cytokines in this set up specially several pioneer labs including Rafi Ahmed, Jonathon Sprent, Charles Surh, Leo Lefrancois and others [10-12] investigated the role of common gamma chain cytokines such as IL-2 and IL-15 and their effect on naïve CD8 T cells phenotype and function.

The common line of evidence that runs through these studies indicates that naïve cells masquerade as memory in the presence of cytokine-rich lymphopenia environment. However, more intriguingly, they showed the necessity of MHC class I in their animal models for this process. For instance, in Murali-Krishna and Ahmed 2000 JI cutting edge study, they observed that naïve cells did not undergo T cell activation and lymphopenia induced proliferation (LIP) upon transfer to irradiated MHC-I knockout (B2m KO) mouse, which hints to the importance and necessity of MHC-I even in the presence of cytokine-rich lymphopenic environment [10]. Further, Cho et al JEM 2007 recapitulate these observations by transferring naïve CD8 T cells into irradiated MHC-I and Tap-1 Knockout mice in the presence of IL-2 complex (IL-2a/IL-2) [11, 13]. These results raise the question whether there is a link between response to gamma chain cytokines and expression of MHC-I. Indeed, Lefrancois lab elegantly showed that adoptive transfer of naïve CD8 T cells into irradiated B2m KO mice did not proliferate compared to WT mice even following injection with IL-15 complex (IL-15/IL-15Ra) [12]. Thus far, the above-mentioned observations give us a clue towards the role of MHC-I-TCR axis possibly by sensitizing naïve cells to be responsive to gamma chain cytokines. Mathew et al., showed the upregulation of CD122 (IL2/15Rb) in polyclonal and LCMV gp33-specific CD8 T cells following LCMV acute infection [14]. On the contrary, gamma-chain cytokines such as IL-2 can reduce TCR threshold and sensitize CD8 T cells to be more responsive to low-binding affinity peptides, which could further explain participation of low-affinity self-antigen specific CD8 T cell clones in an autoimmunity [15]. In summary, these studies demonstrate that although cytokines play a role, MHC-I is still necessary.

5) Acquisition of an activated/memory phenotype by T naive cells requires cell-to-cell contact and is partially inhibited by anti-MHC class-I mAb. This experiments is poorly controlled and must be reproduced using an isotype-control mAb.

5) We thank the reviewer for the suggested experiment and we agree that the usage of isotype controls is an important feature of a well-controlled experiment. However, there are several aspects of using isotype control that made us very cautious to include in our original experimental setup. For instance, off-target effects so-called non-specific binding, which may result in targeting any cell surface protein that we are not aware of, albeit vendors usually claim that the isotype was tested against several target proteins on cell lines, which does not necessarily match the cell type we use in our experiment. Consequently, this scenario might generate a confounding variable that is out of control and might affect our phenotype. Nevertheless, we took the reviewer's suggestion wholehearted and included isotype control in our new experimental setup.

*In these new set of experiments, we thought not only to include an isotype control but also use different concentrations of anti-MHC class I antibody. As shown in **figure 4**, we did reproduce our results using matched isotype antibody with 1:1 ratio co-culture conditions, where we observe a decrease in percentage of naïve CD8 T cells that acquire activated/memory phenotype upon using different concentrations of anti-MHC class I antibody concentrations. These figures were included in the revised version of the manuscript as Fig. 5B-C and Suppl. Fig. 3D.*

6) Related to point 4), it is important to know if MHC class-I molecules bind the TCR or NK receptors: this can be easily assessed using specific mAbs. The binding of MHC to the one or the other receptors has important consequences on the understanding of the molecular basis of cell-to-cell interaction.

6) We thank the reviewer for bringing to our attention to such relevant point. We completely agree with the reviewer's point that we should not ignore other potential ligands that could bind to MHC-I and subsequently regulate the process. Consequently, we thought first to search the literature for studies that discuss the expression of KIR family in T cells. Indeed, Bjorkstrom et al., 2012 Blood manuscript showed that KIRs expression is mainly enriched in terminally differentiated CD8 T cells while naïve CD8 T cells lack KIRs [16]. To confirm these findings in our hands, we examined the expression of KIRs by flow cytometry in T cells. As shown in the **figure 5**, there is minimal expression of KIRs in naïve CD8 T cells cultured alone (Unstimulated) or co-cultured with activated T_{CM} CD8 T cells.

7) Is there need for costimulatory molecules, in addition to MHC class-I molecules. Again, participation of costimulatory molecules may be simply addressed by specific mAbs.

7) This is a very relevant question since naïve T cell responses not only require T cell receptor (TCR)/MHC mediated signal 1 but also costimulatory signal 2 (CD28-CD80/86 axis) as well as soluble mediators such as cytokines (Signal 3). To address this question, we co-cultured naïve CD8 T cells with activated T_{CM} CD8 T cells in the presence of anti-CD80/CD86 ($2\mu\text{g/ml}$) antibodies. As controls, we used isotype at the same concentration. As shown in fig. 6, we did not observe significant change in the frequency of naïve CD8 T cells with activated/memory phenotype between both groups.

8) Do heterologous activated memory T cells induce the same activated/intermediate state in T naïve cells as autologous memory T cells do. This is an important control which must be addressed.

8) We thank the reviewer for his/her suggestion to include such an important control. Indeed, this control will allow us to test the hypothesis whether activated memory T cells are influencing the fate of naïve T cells in an MHC-restricted manner. Consequently, as recommended by the reviewer, we performed new set of experiments by comparing the effect of autologous vs heterologous/allogeneic activated T_{CM} cells on naïve T cells. In these experiments, we co-cultured CTV-labeled naïve CD8 T cells with autologous activated T_{CM} CD8 T cells and compare it to conditions where T_{CM} CD8 T cells are heterologous to naïve CD8 T cells. In the heterologous conditions, we observed an increase in the frequency of naïve CD8 T cells with activated/memory phenotype (red circles) compared to autologous ones. In other donors, we observed the opposite pattern (green circles) and in one donor we did not observe difference between both conditions (black circles) (Fig.7).

In the heterologous co-culture scenario, we speculate that TCMs are presenting nonself-peptides (acting as DCs in mixed lymphocyte reaction [MLR]) to naïve T cells, which results in activation of naïve cells and increase in their frequency (red circles results). In the other scenario (green circles results), probably naïve cells are presenting nonself-peptides to T_{CM} s, which results in rapid activation of T_{CM} s and consequently killing naïve T cells. Hence, there are not enough naïve cells to drive the phenotype into activated/memory T cells. The discrepancy in the phenotype across the donors could be attributed to confounding factors that we are not aware of including various degrees TCR-peptide affinity as well as MHC-disparity between donors. To this end, future studies will be conducted to differentiate between both possibilities.

9) Do the naïve-derived intermediate/memory T cells perform any known functions: this can be inferred from scRNA seq data.

9) We thank the reviewer for bringing up these important analyses. Indeed, we performed function analyses using ingenuity pathway and we observed enrichment of glycolysis function when we compare N3 vs N1 ($p= 0.000000805$ and activation Z-score 2.66). Additionally, cytotoxicity function was enriched when we compare N2 vs N1 ($p= 0.00000000000483$ and activation Z-score 3.07).

10) Authors discuss the possibility (page 14-15) that this novel non-cytotoxic function of CD8 T cells may provide a mechanism for the generation of autoreactive CD8 T cells. I was wondering whether this mechanism may also be relevant for the development of autoimmune diseases.

10) We do agree with the reviewer's point of view. We speculate that these naïve T cells with activated/memory phenotype are autoreactive in nature. Our rationale stems from the possibility that T_{CM} CD8 T cells are cross-talking and presenting self-antigen to naïve CD8 T cells. Future experiments are lined up to test this hypothesis.

11) There are several recent studies (surprisingly none of them is quoted here) that have highlighted that naive T cells are much more heterogeneous than previously thought, and that they harbor diversity in phenotypes, differentiation stages and functions. Please, see the very nice review by Femke van Wijk and colleagues (The full spectrum of human naive T cells. Nat. Rev. Immunol. 2018; 18: 363–73).

11) We thank the reviewer for bringing our attention to this review discussing the heterogeneity of naive T cells. In our analyses, we do observe that a small proportion of naive T cells acquire activated/memory phenotype while majority change to a unique state. These observations could open new research avenues to further define the subset of naive T cells that acquire of activated/memory phenotype.

Reviewer #2 (Remarks to the Author):

A novel non-cytotoxic function of human CD8 T cells

Brief summary and overall impression of the work

This is a well-conducted and well-written piece of original research that examines the question of how activated CD8 memory cells exert control over naïve CD8 T cells. The authors demonstrate that activated CD8 memory cells control activation of naïve CD8 T cells primarily through an MHCI-dependent mechanism. Upon interaction a minority of the naïve cells acquire an activated/memory phenotype, another minor population is still transitioning to its final phenotype(s) at the timepoint analysed and a majority develop into a transcriptionally distinct subset.

This research will be of interest to many immunologists as these interactions between naïve and activated memory cells would be expected to occur in vivo in a variety of situations. Statistics are appropriate throughout. The level of detail provided is sufficient for other researchers to reproduce the work, once minor comments are addressed.

We are very grateful for the reviewer's positive evaluation of our work. Also, we are delighted that the reviewer described our study as well-conducted and well-written. The overall assessment constructively improves the manuscript. Please find below detailed responses to your major and minor points.

Specific comments

Major points

1. Line 29, 30, 62, 265, 272, 280 and others-I do not feel that the data support the idea that 83% are intermediate between naïve and activated memory, as some of these cells have the appearance of taking an entirely different transcriptional path (and would not end up occupying the activated memory space). In other words: I would agree from the UMAP plots and fig4F it appears that the majority of "naïve after co-culture" cells in N3 have started from N1 (were once naïve). However, whilst a minority of these cells appear to be heading towards becoming N2 (i.e. activated TCM), the majority are heading in a different developmental pathway. This is essentially what the authors say in lines 245-250, but the narrative is then confused by the discussion of intermediate cell types that lie between naïve and activated memory. Please could the authors rephrase and discuss where they think the final destination of the majority of subset N3 subset is likely to be?

1. We thank the reviewer for observing the conflict in our manuscript narrative (which was not intentional on our side). We agree with the reviewer point of view since the term intermediate is more suitable for a state in between both naïve and memory, which is not the case in our trajectory analyses. Hence, we updated the text in the revised manuscript at the above-mentioned lines and change it to "hybrid" instead of "intermediate". The reason behind calling N3 as hybrid because their transcription profile is a mixture of naïve and activated memory gene T cell signature. To this end, we are speculating here that since N3 is connected to N2 in a linear branching trajectory, it suggests that cells in N3 state might differentiate to N2 state or vice-versa, which is the most plausible scenario, based on the DEG presented in our manuscript. In conclusion, most likely, a small number of naïve cells become memory (N2) after transiently being in the N3 state while the majority of naïve cells will adopt the N3 transition state and halt at this state. The second scenario could be asymmetric cell division, where naïve T cell divide into proximal and distal cell where N1 give rise to both N2 and N3 simultaneously.

2. scRNAseq: It is not clear in the methods or results how many donors were used and if they were pooled together without hashing (i.e. if the hashing step refers to the different cell types).

- If this represents at least 3 individual donors, with similar percentages in each cell subset, then please just clarify this.
- If it represents one or two donors- then I would not reasonably expect this to be repeated using scRNAseq due to the high costs involved. However, the authors should pick some key markers that represent each of the three neighbourhoods and confirm the results using flow cytometry of at least three individual donors.

2. We thank the reviewer for understanding the high cost of the scRNA-seq experiments and we agree with the reviewer's suggestion. Hence, we picked key markers that can differentiate between the three neighborhoods (please also refer to reviewer#3 minor comments Fig.4). Consequently, based on the scRNA-Seq data, we chose to examine the cell surface protein expression of CD69 as a marker that can differentiate between N1 (true naïve) and N3 (unique hybrid T cell state). Indeed, as shown in fig.8, we observed ~40% of CTV⁺ CD45RO^{neg} population expresses CD69, while ~80% CTV⁺ CD45RO⁺ expresses CD69. Further, as shown in fig.1G-H in the revised manuscript, we were able to differentiate between N1 and N2 (activated TCM) by testing the expression of FasR (CD95) in CTV⁺ CD45RO^{neg} and CD45RO⁺, where N2 express high levels of CD95 compared to CTV⁺ CD45RO^{neg}.

The following text was added to the revised manuscript lines 393-405:

“Since majority of the naïve CD8 T cells that were co-cultured with activated T_{CM} CD8 T cells acquire a unique transcriptional profile i.e., N3 cluster, we thought to validate these changes at a protein level using flow cytometry. As shown in fig. 6D, the upregulation of CD69 transcript could be a good candidate to differentiate between N1 (naïve alone) and N3 (unique T cell state). Hence, we repeated the scRNA-Seq co-culture conditions (activated T_{CM}: TN-3:1) to examine CD69 cell surface protein expression. Our data demonstrated ~40% of CTV⁺ CD45RO^{neg} cell population expresses CD69. Furthermore, we observed a significant upregulation of CD69 in CTV⁺ CD45RO⁺ and CTV⁺ CD45RO^{neg} compared to naïve CD8 T cells in the absence of activated T_{CM} CD8 T cells (Fig. 6E). Additionally, as shown in fig. 1G-H, CD95 protein expression differentiated between CTV⁺ CD45RO^{neg} and CD45RO⁺ cell populations. These data suggest that majority CTV⁺ naïve CD8 T cells co-cultured with activated T_{CM} CD8 T cells upregulate CD69 cell surface protein, which is widely accepted as an activation marker for T cells reflecting an ongoing TCR-dependent responses”.

Minor points

1. line 1: the title could be more descriptive of the function

1. We completely agree with the reviewer suggestion. A new title that includes the function of activated memory T cells has been added to the revised manuscript. **“Activated-memory T cells influence naïve T cell fate: A novel non-cytotoxic function of human CD8 T cells”.**

2. line 57 please could the authors clarify why they have focused on CD8+ cells rather than CD4+ cells

2. We thank the reviewer for bringing this important point to our attention. We included the following paragraph at the beginning of the discussion section lines 416-423.

“Cytotoxic CD8 T cells (CTLs) are classically described as the “serial killers” of the immune system. They play a crucial role in host immune protection against pathogens including viruses, bacteria, parasites, and fungi. Additionally, they can fight tumors if they are not exhausted. Ironically, under certain environmental and genetic conditions they contribute to a wide range of autoimmune diseases e.g., Multiple sclerosis and Rheumatoid arthritis. Furthermore, alloreactive CD8 T cells are considered as one of the main drivers for transplant rejection. Although, the cytotoxic characteristic features of CTLs are well-define, their non-cytotoxic functions have not been studied extensively.”

3. line 69- please clarify that informed consent was obtained from all donors

3. We thank the reviewer for bringing our attention to this ethics matter. We have corrected this issue and it is added to the revised manuscript line 72.

4. line 105 please include cell densities as well as cell number throughout and on this line explain the volume used

4. We completely agree with the reviewer where additional details should be included to this part in the methods section. Further details were added in the revised manuscript lines 140-144.

“Briefly, two wells of T_{CM} CD8 T cells (each at 50K cells/200ul) were activated as described above. Supernatant (total of 300ul) was pooled from both wells followed by two rounds of centrifugation at 450xg to avoid cell contamination. One third of the volume (100ul) was used and added to CTV-labeled naïve CD8 T cells.”

5. line 114- please state the manufacturer or clone id.

5. We have corrected this issue and it is added to the revised manuscript lines 153-158.

“Briefly, CFSE-labeled activated T_{CM} CD8 T cells were co-cultured with CTV-labeled naïve CD8 T cells (1:1) ratio in the presence or absence of anti-MHC class I antibody at a final concentration 2.5, 5, 10 µg/1ml (Ultra-LEAF clone W6/32-Biolegend) at the beginning of the seven days co-culture period. As controls, isotype was added at 2.5 µg/ml concentration. Separate experiments were done at (3:1- memory:naïve) in the presence or absence of anti-MHC class I antibody at a final concentration 5ug/ml”

6. line 117- please state the number of independent replicates and number of donors used (see also major point above)

6. One donor was used in the scRNA-Seq analyses where different cell subsets were sorted and hash-tagged including naïve CD8 T cells cultured alone, activated T_{CM} CD8 T cells cultured alone, and total CTV naïve CD8 T cells (already cultured with activated T_{CM} for 7 days).

7. line 130-please state average number of reads/cell

7. The average number of reads/cell is ~58,000 reads/cell

8. line 133- please explain in more detail the default settings on partek flow, the approach to quality control, demultiplexing and how multiplets were identified.

8. The default settings on partek flow, the approach to quality control, demultiplexing and how multiplets were identified are now explained in more detail in the methods section. The following section was added to the revised manuscript lines 171-184:

scRNAseq and data preprocessing

scRNAseq was performed using 10X Genomics Single Cell 5' solution, version 1, according to the manufacturer's instructions with 4,000 sorted cells loaded for naïve or activated T_{CM} and 9,000 cells for naïve after co-culture. mRNA and hashtag oligos cDNA libraries from pooled samples were sequenced on the NovaSeq6000 Platform (Illumina). Single cell raw matrix files were obtained using the Cell Ranger's pipeline with alignment to the human reference hg38. The raw matrix was then preprocessed and analyzed using Partek Flow v10.0.21.0801 (Partek). Quality control was done using knee point and EmptyDrops [17] to exclude empty droplets. After centered log

ratio (CLR) normalization of the cell hashing count matrix, Hashtag demultiplexing was performed by using an implementation of the algorithm used in Stoeckius et al. 2018 [18] . Multiplets were excluded based on the cell hashing classification and applying an inclusion filter on counts per cell (600-15000) and detected genes per cell (500-4000). Cells with greater than 10% mitochondrial gene expression were excluded to eliminate dead or apoptotic cells, The resulted gene expression matrix was of 9455 cells by 19,327 genes. Single cell gene expression counts were normalized by $\log_2(\text{counts per million} + 1)$.

9. line 140 please explain why UMAP was performed on the top 100 genes of highest variance? In Seurat one would generally use the top 2000 genes for downstream analysis, but perhaps the methodology differs for Partek flow?

9. Please kindly refer to response to reviewer #1's comment#1.

10. line 186, 292 This sentence reads somewhat as if these cells lack cytotoxic capability, whereas actually this is an additional function to control naïve cells, independent of the cytotoxic function. Please could these sentences be clarified?

10. We thank the reviewer for bringing our attention to this important point. To further clarify our conclusions, the following sentence has been added to the revised manuscript lines 239-242:

"Taken together, these results indicate that, in addition to their known cytotoxic functions, activated long-lived memory CD8 T cells (T_{CM} and T_{SCM} subsets) acquire a non-cytotoxic function whereby they shift neighboring naïve CD8 T cells towards an activated/memory phenotype".

11. line 226 please state the values in the text

11. We have corrected this issue and it is added to the revised manuscript.

12. line 312 in addition to MHCI interaction and minor contributions from soluble factors, might other cell contact dependent interactions play a minor role in the cross talk between naïve and activated memory cells? Please discuss.

12. Please refer to Reveiwer#1 point#7.

13. Throughout the figures- I do not feel it is helpful to put the fold change above the asterisks

13. We did remove the fold change above the asterisks from all the figures as suggested by the reviewer.

14. Figure 2B- the CFSE graphs should be described more fully in the text

14. We thank the reviewer for this valid point and we agree regarding the lack of clarity in this section. Hence, we included more details that are added to the revised manuscript lines 257-268.

"Since we observed that the activated T_{CM} and T_{SCM} subsets had the highest proliferative capacity and the most pronounced effect in our co-culture experiments (Fig. 1C and 2A), we asked whether the acquisition of an

activated/memory phenotype by naïve T cells correlates with memory T cell proliferation. To test this hypothesis, we performed simple linear regression test using Pearson correlation coefficient analyses to draw a relationship between two variables: (1) percentage of CD45RO⁺ CTV⁺ CD8 T cells and (2) percent dividing memory T cells within the total memory population following coculture with naïve CD8 T cells (**Fig. 2C**). In our analyses, we found a direct correlation between both variables ($R^2 = 0.36$, $p = 0.001$) (**Fig. 2D**), in which an increase in the proliferation of memory CD8 T cells specially T_{CM} is associated with increase in the frequency of naïve CD8 T cells with acquired activated/memory phenotype.”

15. Figure 2B/D- I appreciate that this CFSE/CTV experiment had to run for 7 days and therefore the divisions are not as sharp as would be seen at an earlier timepoint. However, some staining looks very blurry/low. Please could a supplementary figure be provided that confirms uniform staining at day 0 and/or well-defined division around day 3?

15- We thank the reviewer for bringing our attention to this issue and we agree regarding the blurriness of the figure when we use contour plots. Hence, we switched to pseudocolor smooth plots to enhance the clarity of the plots as shown in the figure below (**Fig. 9A**). Additionally, as suggested by the reviewer, we performed new experiment to show well defined cell division at earlier time points. In this experiment, we did an overnight stimulation of T_{CM} followed by beads removal. We then measured proliferation at three time points (Day 0, 3 and 6). As shown in the figure 9B, there was no obvious proliferation at day 0 (18 hrs following beads stimulation) while at Day 3 T_{CM} cells start to proliferate and by day 6 we saw complete proliferation. We included these results in Suppl. Fig.2A and described in the text of Fig.2 results section.

16. Figure 4A- is confusing, some additional arrows would clarify where the bottom pool of cells come from?

16. We apologize for any lack of clarity in fig4A (fig.6A in the updated revised manuscript). Indeed, we added an arrow to show that the bottom population is coming from co-culture of naïve with activated T_{CM} CD8 T cells.

17. Figure 4B/C further on the point about only using 100 genes for the UMAP projection (and without knowing the default setting for Partek flow), these UMAP plots appear to be somewhat underclustered? Altering the analysis parameters may allow more intermediate populations to be revealed (and possibly give the option of visualising the trajectory analysis on the UMAP plot)?

17. Please refer to reviewer #1's comment#1. The UMAP settings and the default settings for data preprocessing are now listed in the methods section. Monocle 2 is a separate algorithm that does not utilize UMAP projection for dimensionality reduction. Hence the trajectory analysis cannot be overlaid on the UMAP projection.

Stephanie J Hanna

Reviewer #3 (Remarks to the Author):

Review:

This is an interesting and well-presented study which highlights the potential for human memory CD8+ T cells to directly activate their naïve counterparts, driving them towards a memory phenotype via autologous MHC class I interactions and in the absence of a conventional APC priming event. The series of in vitro experiments have been well-designed and clearly described and includes imagestream and transcriptomic analysis of the activated naïve T cells. The paper does have the potential to add to our understanding of T cell biology, however, I do have some comments that the authors would need to address.

We are very appreciative for the reviewer's well thought and to the point comments that actually drives the study in the right direction. We are very happy that the reviewer found our study interesting, well-presented and has the potential to add to our understanding of T cell biology. Please kindly refer to the below point-by-point response to your comments/suggestions.

General comments:

1. Although it is clear that cell contact is required to initiate the activation of autologous naïve T cells in co-culture, the potential downstream role of IL2 in driving naïve CD8 T cells towards a memory phenotype, which has already been described in the literature, should be acknowledged/discussed.

1. This is a very relevant point to our study and we completely agree with reviewer that we should take it in consideration and understand the role of IL-2 and other gamma chain cytokines in our in vitro system. Indeed, several studies by Rafi, Sprent, Surh, Lefrancois, and others demonstrated the importance of common gamma chain cytokines such as IL-2 and IL-15 and their effect on naïve CD8 T cells [10-12].

The common thread in these studies showed that naïve T cells acquire memory-like phenotype and function in a lymphopenic environment which is rich in cytokines. Intriguingly, they also showed the importance of MHC-I in driving this phenotype using B2m KO (lack of MHC-I) mice. For example, in a JI cutting edge study by Rafi's lab, the authors showed that naïve cells cannot proliferate or get activated following adoptive transfer into irradiated B2m KO mice compared to WT mice. Later on, similar observations have been reproduced by Sprent and Lefrancois labs even following injection of IL-15 complex (IL-15/IL-15Ra) or IL-2 complex into irradiated B2m KO mice [10-12].

These observations give us a clue towards the role of MHC-I-TCR axis in making naïve cells responsive to gamma chain cytokines. Indeed, Mathew et al. demonstrated the upregulation of CD122 (IL2/15Rb- receptor subunit common between IL-2 and IL-15) in polyclonal and LCMV gp33-specific CD8 T cells following LCMV acute infection [14]. On the contrary, gamma-chain cytokines such as IL-2 can reduce TCR threshold and sensitize CD8 T cells to be more responsive to low-binding affinity peptides, which could further explain participation of low-affinity self-antigen specific CD8 T cell clones in an autoimmunity [15]. In summary, these studies demonstrate that although cytokine-rich environment play a role, MHC-I is still necessary for the process (please also refer to Reviewer#1 point#4 pg.6 for a similar response).

2. The contact-dependent mechanism has been attributed here to autologous MHC-Class I/TCR interaction between the activated CD8+ memory and naïve CD8+ T cells but there is no clear explanation of why this does not occur in the absence of prior CD8 memory cell activation (non-activated TCM will still express MHC class I). Can the authors comment on this? Presumably there are additional factors activated by the polyclonal stimulation of memory cells prior to co-culture? Can the effect also be blocked with anti-IL2 or by blocking costimulatory receptor signalling?

2. We thank the reviewer for such an elegant observation and stimulating discussion which gives us insights to deeply understand this interesting phenomenon.

To answer the above-mentioned question, we first measured the expression of MHC-I in TCM CD8 T cells pre- and post-TCR stimulation. Indeed, we observed an increase in the MFI of MHC-I in TCM CD8 T cells post-TCR stimulation (6 days post removal of 18hrs beads stimulation) compared to unstimulated controls (Fig.10). The upregulation of MHC-I in stimulated TCM CD8 might be one of the primary factors and represents signal 1 that probably initiates and contribute for the generation of naïve with acquired memory phenotype following co-culture. Nevertheless, as mentioned by the reviewer, there could be parallel events that is happening alongside upregulation of MHC-I. Since it is expected that a wide range of cytokines (signal 3) including gamma chain cytokines such as IL-2 might be upregulated upon TCR/CD28 stimulation, we measured IL-2 in the supernatant of TCM CD8 T cells at day7 following beads stimulation (18hrs beads stimulation). As shown in Fig.10, we observed an increase in the secretion of IL-2 in the supernatant of stimulated TCM CD8 T cells with an average of 1300 pg/ml.

Fig. 10: Bar-graphs depicting MHC class-I MFI and IL-2 expression in TCM following anti-CD3/28 beads stimulation.

Fig. 11: FACS plots and bar-graph showing the effect of IL-2 on naïve CD8 T cells in the presence of unstimulated TCM CD8 T cells.

To test if we can recapitulate the phenotype in the presence of IL-2. We added IL-2 cytokine exogenously using the average conc. from figure 10 to the co-culture of naïve with unstim. TCM CD8 T cells. As shown in **figure 11**, to our surprise, we did not observe the phenotype, which hints that

other homeostatic cytokine(s) is working with IL-2 i.e., IL-15 to drive this phenotype. Our transwell data (**Figure 12**) support a cell contact mechanism and IL-15 is known to be trans-presented by IL-15Ra to bind to IL-2/IL15Rb (CD122)-CD132 (common gamma chain) receptor complex on naïve T cells, we hypothesize that IL-15Ra could play a role through trans-presenting IL-15 to naïve T cells where MHC-I/TCR interaction upregulate CD122 on naïve T cells making them sensitive to IL-15. Future experiments will be performed to test this hypothesis. Also, we did block signal 2 and 3 as suggested by the reviewer. Please refer to Reviewer#1 point#4 (pg3-5) and #7 (pg.8) for further details.

Fig. 12: FACS plots and bar-graph comparing co-culture versus transwell conditions and their effect on acquisition of activated/memory phenotype on naïve T cells.

3. The authors claim, based on transcriptomic data, that this is a potential non-cytotoxic role for CD8+ T cells but lack of cytotoxicity has not been formally demonstrated at the protein or functional level.

3. This is a very relevant point that we should clarify in our study. Indeed, we rephrase the text as follows in the revised manuscript lines 240-242.

“Taken together, these results indicate that, in addition to their known cytotoxic functions, activated long-lived memory CD8 T cells (T_{CM} and T_{SCM} subsets) acquire a non-cytotoxic function whereby they shift neighboring naïve CD8 T cells towards an activated/memory phenotype”. Further to confirm our point, we did observe upregulation in gene expression of effector and cytotoxic genes in N2 cluster (activated TCM CD8 T cells) compared to naïve alone (**Fig. 6D at the revised manuscript**). Also, we performed function analyses using ingenuity pathway and we observed enrichment of cytotoxicity function when we compare N2 vs N1 ($p= 0.00000000000483$ and activation Z-score 3.07). Also please refer to Reviewer#1 point#9 for a similar response.

4. In general there is a need to be clear in the figure legends about the number of biological replicates/samples that data represent.

4. We thank the reviewer for pointing out this issue. Indeed, we had corrected it in the revised manuscript.

Minor comments:

Figure 1A

If plate magnet is used to remove beads, is the subsequent co-culture performed in the same media? If so, how can the authors control for contribution of IL2 or other cytokines from the stimulated memory Teff population?

Figure 1A: We are very glad that the reviewer asks this question giving us the opportunity to further clarify our approach and think more about confounding factors that could contribute to the phenotype. According to the in vitro co-culture system we designed in this study, we did collect the overnight supernatant following stimulation of T_{CM} CD8 T cells and add to the unmanipulated naïve CD8 T cells.

*To control for the contribution of IL-2 or other cytokines in our system, as discussed above in general comments point#2, we thought first to measure IL-2 protein levels in the supernatant. Indeed, we showed that activated T_{CM} CD8 T cells secrete IL-2 compared to unstimulated T cells (**Fig.9 point#2**). The rationale behind measuring IL-2 in the supernatant is to determine the actual levels of IL-2 that could be secreted by activated T_{CM} CD8 T cells so we can then use this conc. in our co-culture system. Hence, this approach will guide us to an educated guess of IL-2 conc. instead of using supraphysiological conc. previously documented in the literature [19, 20]. These high conc. does not reflect physiological levels of IL-2 in a lymphopenic environment and could induce non-specific proliferation of T cells. Indeed, Cho et al. showed that the serum levels of IL-2 in CD122ko mice are significantly higher ~200 pg/ml compared to WT mice [11]. Additionally, the levels of other gamma chain cytokines such as IL-7 and IL-15 were at the magnitude of 20-100pg/ml in lymphopenia patients [21], which again suggest that the conc. of gamma chain cytokines including IL-2 should be wisely chosen at least for the purpose of our study.*

*We next activated CFSE-labeled T_{CM} CD8 T cells then co-culture them with CTV-labeled naïve CD8 T cells in the presence of anti-human IL-2 antibody. As controls, we used isotype antibody at the same concentration. Following 6 days of the co-culture, we examined the proliferation capacity of CFSE-labeled T_{CM} CD8 T cells as well as the phenotype of CTV-labeled naïve CD8 T cells. We observed a significant decrease in the proliferation capacity of memory T cells and concurrently the frequency of naïve CD8 T cells with an activated/memory phenotype was much less compared to isotype controls. These results demonstrate that IL-2 plays an indirect role in the acquisition of activated/memory phenotype by naïve CD8 T cells via proliferation of memory CD8 T cells (**Please refer to Reviewer#1 point#4 pg3-5 Fig. 3E-F**). (Also added to the manuscript as Suppl. figure 3C-D).*

*However, our approach still did not address whether blocking IL-2 will have a direct effect on naïve CD8 T cells to acquire activated/memory phenotype independent of the proliferation of memory CD8 T cells. Hence, we added rhIL-2 dose at 1000 pg/ml conc. as well as 10x more to the co-culture of naïve and unstimulated T_{CM} CD8 T cells (**Fig. 11**). To our surprise, we did not observe change in the phenotype of naïve T cells compared to naïve T cells co-cultured with stimulated T_{CM} CD8 T cells. These results suggest that MHC-TCR axis (signal 1) and probably other soluble factor(s) could be the early events required to initiate the proliferation of T_{CM} CD8 T cells and hence acquisition of activated/memory phenotype by naïve CD8 T cells. Thus far, our data reveal a cause-and-effect relationship between acquisition of activated/memory phenotype by naïve CD8 T cells and the proliferation capacity of activated memory T_{CM} CD8 T cells.*

Figure 1B

The authors correctly do not gate CTV- naïve T cells to avoid including activated/proliferated memory CD8 T cells which might overlap with the CTV- population. However, this would not control for auto fluorescence of the activated/proliferated memory T cells in the CTV channel. An additional control of stim CFSE labelled memory cells alone (with no TN) would control for this.

Figure 1B: We thank the reviewer for such an important point and we do agree with the reviewer's suggestion. Please refer to Reviewer#2 point#15 Figure 9B. Suppl. Fig. 2A was added to the revised manuscript.

Figure 1D

The imagestream data is an excellent addition and helpful in demonstrating a clear surface upregulation of CD45RO on a CTV+ (thus previously naïve) single cell. However, what is this figure representative of (i.e. how many biological/technical replicates)?

Figure 1D: We are very glad that the reviewer mentioned that the image stream data is an excellent addition to our manuscript and we thank the reviewer for such a nice compliment. Fig 1D represents one event that happens in one biological donor (TN + activated TCM). We included another event from the same donor as shown in the figure below.

Figure 2

FigD needs further clarification of gating since it is difficult in this bottom left quadrant to be completely clear what is divided memory CFSE+ (now CFSE-) cells versus proliferated naïve (CTV+ now CTV-) cells and it is not clear how the % CD45RO+CTV+ Naïve T cells in Fig 3E have been derived because the gating of naïve/CTV+ T cells is not shown. Additional timepoints should be considered here - gating of CFSE dividing cells would be much clearer for example at day 3-5 when memory cells are divided but not completely CFSE negative.

Figure 2D: Please refer to Reviewer#2 point#15 Figure 9A

Figure 3

Fig 3D Can the authors comment on the direction of blocking since presumably pan-class I antibody could potentially block MHC Class I on both the naïve and memory T cells.

As per my earlier comment, since unstimulated memory CD8 T cells also express MHC Class I can the authors comment on the need for stimulation of the memory cells in inducing naïve T cell activation?

Figure 3: We thank the reviewer for bringing up this important point regarding the directionality of the blocking. Indeed, in our new experiments (Reviewer#1 point#5) we added anti-MHC class I antibody during the overnight stimulation of TCM CD8 T cells assuming blocking will occur during the overnight

incubation. However, it is not granted that the antibody will block MHC class I on naïve T cells as well since we collect the supernatant along with activated TCM CD8 T cells to add on unmanipulated naïve CD8 T cells. The best approach will be usage of CRISPR/Cas9 technology to knock out B2m specifically in activated TCM CD8 T cells then co-culture with naïve T cells. Future experiments are planned to apply this approach.

Figure 4

If 85% of the naïve T cells undergo a transcriptomic shift to N3 can some of these changes be validated at the protein level? For example, upregulation of CD69 would be easy to demonstrate. Lactate can be used as a readout of glycolysis. As per my earlier comment the authors should be clear that non-cytotoxic function has not been demonstrated functionally but rather transcriptomically.

Figure 4: We thank the reviewer for the nice suggestion and the proposed experiments to enhance our manuscript. Consequently, we examined the cell surface protein expression of CD69 as a marker that can differentiate between N1 (true naïve) and N3 (unique T cell state). As shown in the fig.13, we observed ~40% of CTV+ CD45ROneg population expresses CD69.

The following text has been added to the revised version of the manuscript lines 393-405.

“Since majority of the naïve CD8 T cells that were co-cultured with activated TCM CD8 T cells acquire a unique transcriptional profile i.e., N3 cluster, we thought to validate these changes at a protein level using flow cytometry. As shown in fig. 6D, the upregulation of CD69 transcript could be a good candidate to differentiate between N1 (naïve alone) and N3 (unique T cell state). Hence, we repeated the scRNA-Seq co-culture conditions (activated TCM : TN-3:1) to examine CD69 cell surface protein expression. Our data demonstrated ~40% of CTV+ CD45ROneg cell population expresses CD69. Furthermore, we observed a significant upregulation of CD69 in CTV+ CD45RO+ and CTV+ CD45ROneg compared to naïve CD8 T cells in the absence of activated TCM CD8 T cells (Fig. 6E). Additionally, as shown in fig. 1G-H, CD95 protein expression differentiated between CTV+

CD45RO^{neg} and CD45RO⁺ cell populations. These data suggest that majority CTV⁺ naïve CD8 T cells co-cultured with activated T_{CM} CD8 T cells upregulate CD69 cell surface protein, which is widely accepted as an activation marker for T cells reflecting an ongoing TCR-dependent responses.

References:

1. Wang, Z., et al., *Single-cell RNA sequencing of peripheral blood mononuclear cells from acute Kawasaki disease patients*. Nat Commun, 2021. **12**(1): p. 5444.
2. Borel, J.F., et al., *Effects of the new anti-lymphocytic peptide cyclosporin A in animals*. Immunology, 1977. **32**(6): p. 1017-25.
3. Borel, J.F. and D. Wiesinger, *Studies on the mechanism of action of cyclosporin A [proceedings]*. Br J Pharmacol, 1979. **66**(1): p. 66P-67P.
4. Cantrell, D.A. and K.A. Smith, *Transient expression of interleukin 2 receptors. Consequences for T cell growth*. J Exp Med, 1983. **158**(6): p. 1895-911.
5. Stern, J.B. and K.A. Smith, *Interleukin-2 induction of T-cell G1 progression and c-myc expression*. Science, 1986. **233**(4760): p. 203-6.
6. Kronke, M., et al., *Cyclosporin A inhibits T-cell growth factor gene expression at the level of mRNA transcription*. Proc Natl Acad Sci U S A, 1984. **81**(16): p. 5214-8.
7. Bunjes, D., et al., *Cyclosporin A mediates immunosuppression of primary cytotoxic T cell responses by impairing the release of interleukin 1 and interleukin 2*. Eur J Immunol, 1981. **11**(8): p. 657-61.
8. Orosz, C.G., et al., *Analysis of cloned T cell function. II. Differential blockade of various cloned T cell functions by cyclosporine*. Transplantation, 1983. **36**(6): p. 706-11.
9. Calder, V.L., et al., *Effects of cyclosporin A on expression of IL-2 and IL-2 receptors in normal and multiple sclerosis patients*. Clin Exp Immunol, 1987. **70**(3): p. 570-7.
10. Murali-Krishna, K. and R. Ahmed, *Cutting edge: naive T cells masquerading as memory cells*. J Immunol, 2000. **165**(4): p. 1733-7.
11. Cho, J.H., et al., *An intense form of homeostatic proliferation of naive CD8+ cells driven by IL-2*. J Exp Med, 2007. **204**(8): p. 1787-801.
12. Stoklasek, T.A., et al., *MHC class I and TCR avidity control the CD8 T cell response to IL-15/IL-15Ralpha complex*. J Immunol, 2010. **185**(11): p. 6857-65.
13. Sprent, J., et al., *T cell homeostasis*. Immunol Cell Biol, 2008. **86**(4): p. 312-9.
14. Mathews, D.V., et al., *CD122 signaling in CD8+ memory T cells drives costimulation-independent rejection*. J Clin Invest, 2018. **128**(10): p. 4557-4572.
15. Au-Yeung, B.B., et al., *IL-2 Modulates the TCR Signaling Threshold for CD8 but Not CD4 T Cell Proliferation on a Single-Cell Level*. J Immunol, 2017. **198**(6): p. 2445-2456.
16. Bjorkstrom, N.K., et al., *CD8 T cells express randomly selected KIRs with distinct specificities compared with NK cells*. Blood, 2012. **120**(17): p. 3455-65.
17. Lun, A.T.L., et al., *EmptyDrops: distinguishing cells from empty droplets in droplet-based single-cell RNA sequencing data*. Genome Biol, 2019. **20**(1): p. 63.
18. Stoeckius, M., et al., *Cell Hashing with barcoded antibodies enables multiplexing and doublet detection for single cell genomics*. Genome Biol, 2018. **19**(1): p. 224.
19. Liu, Y., et al., *Effects of interleukin-2 concentration and administration method on proliferation and function of cytokine-induced killer cells*. Transl Cancer Res, 2021. **10**(9): p. 3930-3938.
20. Teague, R.M., et al., *Proliferation and differentiation of CD8+ T cells in the absence of IL-2/15 receptor beta-chain expression or STAT5 activation*. J Immunol, 2004. **173**(5): p. 3131-9.
21. Kielsen, K., et al., *IL-7 and IL-15 Levels Reflect the Degree of T Cell Depletion during Lymphopenia and Are Associated with an Expansion of Effector Memory T Cells after Pediatric Hematopoietic Stem Cell Transplantation*. J Immunol, 2021. **206**(12): p. 2828-2838.

REVIEWERS' COMMENTS:

Reviewer #1 (Remarks to the Author):

I do not have any additional comment to authors.

Reviewer #2 (Remarks to the Author):

Thank you for your rebuttal letter, which has addressed all of my initial concerns. My only further comment would be in response to

9. line 140 please explain why UMAP was performed on the top 100 genes of highest variance? In Seurat one would generally use the top 2000 genes for downstream analysis, but perhaps the methodology differs for Partek flow?

9. Please kindly refer to response to reviewer #1's comment#1.

In this response you have included a UMAP based on the top 2000 genes. I think it would be useful to have this as a supplementary figure, as I think many readers will have similar questions to myself and reviewer 1 about using only 100 genes. Having this as a supplementary figure would reassure them.

Reviewer #3 (Remarks to the Author):

The authors have done well to address all the concerns and comments made in response to review of their article. This has included a substantial amount of new data and rewording of parts of the manuscript, as well as helpful clarification on other points and explanations where further work was not possible at this time. I would now be very happy to recommend this article for publication in Communications Biology

We thought to take the opportunity and thank all the reviewers for taking the time to review our manuscript and provide their insightful comments/suggestions. We also would like to thank the editor very much for her dedicated time and help along the whole process.

Reviewer #1 (Remarks to the Author):

I do not have any additional comment to authors.

We thank again the reviewer for the insightful comments and constructive assessment of our manuscript.

Reviewer #2 (Remarks to the Author):

Thank you for your rebuttal letter, which has addressed all of my initial concerns. My only further comment would be in response to

9. line 140 please explain why UMAP was performed on the top 100 genes of highest variance? In Seurat one would generally use the top 2000 genes for downstream analysis, but perhaps the methodology differs for Partek flow?

9. Please kindly refer to response to reviewer #1's comment#1.

In this response you have included a UMAP based on the top 2000 genes. I think it would be useful to have this as a supplementary figure, as I think many readers will have similar questions to myself and reviewer 1 about using only 100 genes. Having this as a supplementary figure would reassure them.

We are very appreciative for the reviewer's well thought comments and suggestions. Indeed, we included the UMAP figure as suppl. Fig. 2b in the revised manuscript. The methods section is updated accordingly.

Reviewer #3 (Remarks to the Author):

The authors have done well to address all the concerns and comments made in response to review of their article. This has included a substantial amount of new data and rewording of parts of the manuscript, as well as helpful clarification on other points and explanations where further work was not possible at this time. I would now be very happy to recommend this article for publication in Communications Biology

We are very grateful for the reviewer's comments and suggested experiments that helped us significantly in the revision process. We are delighted that the reviewer recommended our study for publication in Communications Biology and mentioned that we were able to address/clarify all the comments and concerns raised by the reviewers.